# Earth system models underestimate carbon fixation by plants in the high latitudes

Alexander J. Winkler [1,2], Ranga B. Myneni[1,3], Georgii A. Alexandrov[4] & Victor Brovkin [1]

Most Earth system models agree that land will continue to store carbon due to the physiological effects of rising $CO_2$ concentration and climatic changes favoring plant growth in temperature-limited regions. But they largely disagree on the amount of carbon uptake. The historical $CO_2$ increase has resulted in enhanced photosynthetic carbon fixation (Gross Primary Production, GPP), as can be evidenced from atmospheric $CO_2$ concentration and satellite leaf area index measurements. Here, we use leaf area sensitivity to ambient $CO_2$ from the past 36 years of satellite measurements to obtain an Emergent Constraint (EC) estimate of GPP enhancement in the northern high latitudes at two-times the pre-industrial $CO_2$ concentration ($3.4 \pm 0.2$ Pg C yr$^{-1}$). We derive three independent comparable estimates from $CO_2$ measurements and atmospheric inversions. Our EC estimate is 60% larger than the conventionally used multi-model average (44% higher at the global scale). This suggests that most models largely underestimate photosynthetic carbon fixation and therefore likely overestimate future atmospheric $CO_2$ abundance and ensuing climate change, though not proportionately.

[1] Max-Planck-Institute for Meteorology, Bundesstrasse 53, 20146 Hamburg, Germany. [2] International Max-Planck Research School for Earth System Modeling, Bundesstrasse 53, 20146 Hamburg, Germany. [3] Department of Earth and Environment, Boston University, Boston, MA 02215, USA. [4] A.M. Obukhov Institute of Atmospheric Physics, Pyzhyovskiy Pereulok 3, Moscow 119017, Russia. Correspondence and requests for materials should be addressed to A.J.W. (email: alexander.winkler@mpimet.mpg.de)

Predicting climate change requires knowing how much of the emitted $CO_2$ (currently ~40 Pg $CO_2$ yr$^{-1}$) will remain in the atmosphere (~46%) and how much will be stored in the oceans (~24%) and lands (~30%)[1]. Earth system models (ESM) show a large spread in projected increase of terrestrial photosynthetic carbon fixation (GPP)[2–6] and are thought to overestimate current estimates[5,7], although the latter is also subject of debate[5,8–11]. Historical increase of atmospheric $CO_2$ concentration, from 280 to current 400 ppm, has resulted in enhanced GPP due to its radiative[12] and physiological[13,14] effects, which is indirectly evident in amplified seasonal swings of atmospheric $CO_2$ concentration[15–17] and large scale increase in summer time green leaf area[18–20]. Thus, these observables, expressed as sensitivities to ambient $CO_2$ concentration, might serve as predictors of changes in GPP[21–24] and help to reduce uncertainty in multi-model projections of terrestrial carbon cycle entities.

This study is focused on the northern high latitudes (NHL, north of 60°N) where significant and linked changes in climate[25] and vegetation[15] have been observed in the past 3–4 decades: 52% of the vegetated lands show statistically significant greening trends over the 36-year record of satellite observations[26] (1981–2016, Methods), while only 12% show browning trends, mostly in the North American boreal forests due to disturbances[27] (Fig. 1). We therefore hypothesize that the greening sensitivity (i.e., leaf area index, LAI, changes in response to changes in the driver variables) inferred from the historical period of $CO_2$ increase can be used to obtain a constrained estimate[23] of future GPP enhancement from both the radiative and physiological effects (Supplementary Fig. 1).

State-of-the-art fully coupled carbon-climate ESMs vary in their representation of many key processes, e.g., vegetation dynamics, carbon–nitrogen interactions, physiological effects of $CO_2$ increase, climate sensitivity, etc. This results in divergent trajectories of evolution of the 21st century carbon cycle[4–6]. To capture this variation, we use two sets of simulations[28] available from seven ESMs[23] from the Coupled Model Intercomparison Project Phase 5 (CMIP5)—one with historical forcings including anthropogenic $CO_2$ emissions for the period 1850–2005 and the second with idealized forcing (1% $CO_2$ increase per year, compounded annually, starting from a pre-industrial value of 284 ppm until quadrupling). In our analyses, the magnitude of the physiological effect is represented by the $CO_2$ concentration and the radiative effect by growing degree days (GDD0, > 0 °C, Methods) as plant growth in NHL is principally limited by the growing season temperature[12]. Leaf area changes can be represented either by changes in annual maximum LAI (LAI$_{max}$)[29] or growing season average LAI—we use the former because of its ease and unambiguity, as the latter requires quantifying the start- and end-dates of the growing season, something that is difficult to do accurately in NHL[30] with the low-resolution model data.

Here, we apply the concept of Emergent Constraints (EC) to reduce uncertainty in multi-model projections of GPP using historical simulations and satellite observations of LAI focusing on NHL. We find that the EC estimate is 60% larger than the commonly accepted multi-model mean value, in line with a recent study that assessed the impact of physiological effects of higher $CO_2$ concentration on GPP of northern hemispheric extra-tropical vegetation[23]. Detailed independent analyses of in-situ $CO_2$ measurements and atmospheric inversions imbue confidence in our conclusions. Our central finding is, the effect of ambient $CO_2$ concentration on terrestrial photosynthesis is larger than previously thought, and thus, has important implications for future carbon cycle and climate.

## Results

**Large inter-model spread in greening sensitivity.** The enhancement in NHL greenness throughout the observational period relates linearly to both increasing quantities, GDD0 and $CO_2$ concentration, in general agreement between models and observations[15,19,31]. To avoid redundancy from co-linearity between the two driver variables, but retain their underlying time-trend and interannual variability (Supplementary Table 1), we use the dominant mode from a principal component analysis (PCA) of $CO_2$ and GDD0 as the proxy driver (denoted ω) in subsequent analysis (Methods). Expressed in this compact form, greenness level (Fig. 2a) as well as greening sensitivity to ω (hereafter greening sensitivity, Fig. 2b) span a wide range across the multi-model ensemble. All models with low greenness levels (LAI$_{max}$ < 0.75 m$^2$ m$^{-2}$) tend to simulate low greening sensitivities (< 0.015 m$^2$ m$^{-2}$ LAI$_{max}$ per 1 unit ω), relative to observations. These models (NorESM1-ME, CESM1-BGC, and CanESM2) lack a representation of dynamic vegetation, i.e., do not allow plant functional type shifts in response to changing simulated climate, and/or show overly strong nitrogen limitations on plant growth and thus fail to capture GPP enhancement and its re-investment in green leaf area (Supplementary Table 2). The other four models behave randomly—some reproduce observed greenness levels (LAI$_{max}$ ~1.7 m$^2$ m$^{-2}$) but not the greening sensitivities (~0.045 m$^2$ m$^{-2}$ LAI$_{max}$ per 1 unit ω) and the others vice versa. Whether this is because these four models in common lack carbon–nitrogen interactions, or are missing some other key processes, is not known[31], but the end result is a large range in model simulated greening sensitivity (hereafter LAI$_{max}$ sensitivity), during the historical period (0.022–0.075 m$^2$ m$^{-2}$ LAI$_{max}$ per 1 unit ω).

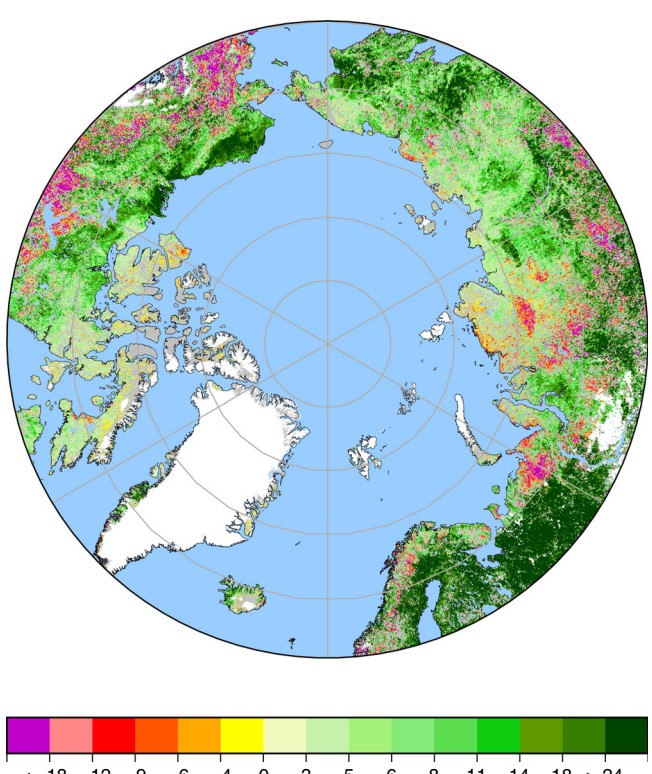

**Fig. 1** Greening (LAI increase) and browning trends during 1981–2016 in the northern high latitudes. Statistically significant (Mann–Kendall test, $p < 0.1$) trends in summer (June–August) average LAI are color coded. Non-significant changes are shown in gray. White areas depict ice sheets or barren land. Details of the LAI data set are provided in Methods. The figure was created using the cartographic python library Cartopy (Release: 0.16.0)

Trend in annual summer (JJA) LAI, $10^{-2}$ m$^2$ m$^{-2}$ decade$^{-1}$

< −18  −12  −9  −6  −4  0  3  5  6  8  11  14  18  > 24

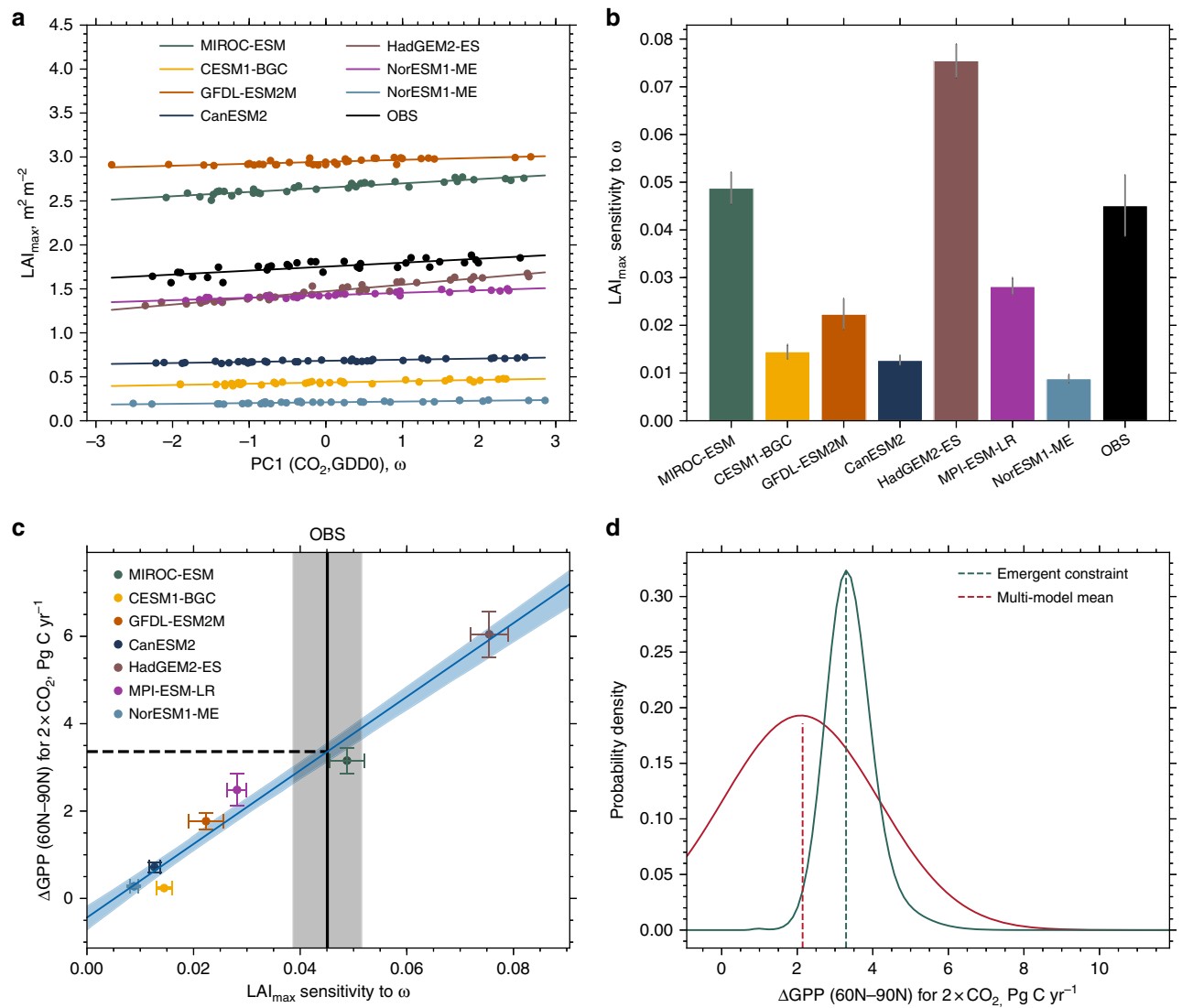

**Fig. 2** CMIP5 ensemble mean considerably underestimates absolute increase of GPP for a doubling of pre-industrial atmospheric $CO_2$ concentration ($2 \times CO_2$). **a** Observations (black) and CMIP5 historical simulations (colors) of the first principal component of annual mean atmospheric $CO_2$ and annual growing degree days above 0 °C ($\omega$) versus the annual $LAI_{max}$. All quantities are area weighted and spatially averaged for NHL (60°N–90°N). **b** Bar chart showing the corresponding slopes of the best linear fits, where the gray bar at the top indicates the standard error. Linear trends are derived for the period 1982–2016 for observations and 1971–2005 for model simulations, maximizing the overlap and sample size. **c** Linear relationship between the sensitivity of annual $LAI_{max}$ to $\omega$ (x axis) and the absolute increase of high-latitude GPP at $2 \times CO_2$. Each model is represented by an individually colored marker with error bars indicating one standard deviation (y axis) and standard error (x axis). The black solid line shows observed sensitivity, where the gray shading indicates the respective standard error. The blue line shows the best linear fit across the CMIP5 ensemble including the 68% confidence interval estimated by bootstrapping (blue shading; Methods). The intersection of the blue and black line gives the Emergent Constraint on $\Delta$GPP at $2 \times CO_2$ (dashed black line). **d** Probability density functions resulting from Emergent Constraint (blue) and CMIP5 ensemble mean estimates (red, assuming Gaussian distribution). Details in Methods

**Emergent Constraint on projected increase of GPP.** What is known, however, is the strong linear relationship between modeled contemporaneous changes in $LAI_{max}$ and GPP arising from the combined radiative and physiological effects of $CO_2$ enrichment in the range $1 \times CO_2$ to $2 \times CO_2$ (Supplementary Figs. 2 and 3). As a result, models with low $LAI_{max}$ sensitivity (Fig. 2b) project lower $\Delta$GPP for a given increment of $CO_2$ concentration, and vice versa. Thus, the large variation in modeled historical $LAI_{max}$ sensitivities (Fig. 2b) linearly maps to variation in $\Delta$GPP at $2 \times CO_2$ (Fig. 2c; $r = 0.98$, $P = 0.0001$), with the consequence that the uncertainty of the multi-model mean $\Delta$GPP is large enough to undermine its value—e.g., $2.1 \pm 1.91$ Pg C yr$^{-1}$ for $2 \times CO_2$ in NHL. This linear relation in inter-model variation

between $\Delta$GPP at $2 \times CO_2$ and historical $LAI_{max}$ sensitivities allows using the observed sensitivity as an EC on GPP estimation at $2 \times CO_2$. Moreover, the probability contours about the best linear fit together with the uncertainty of observed sensitivity (blue and gray shadings in Fig. 2c) allow a robust characterization of the constrained estimate[23], namely $3.4 \pm 0.2$ Pg C yr$^{-1}$ for $2 \times CO_2$ in NHL (Fig. 2d). This EC estimate is 60% larger than the multi-model mean value. Wenzel et al.[23] reported a similar result for NHL ($37 \pm 9\%$ versus 20–25% for relative GPP increase at $2 \times CO_2$) and a somewhat smaller number for the extra-tropical vegetation in the northern hemisphere, both for the physiological effect only (Supplementary Fig. 4 shows that the radiative and physiological effects each contribute about half of the total GPP

enhancement). Together, these results indicate that most models are largely underestimating photosynthetic carbon fixation, which is in contrast to previous studies[5,7] that suggested an over-sensitivity of ESMs to atmospheric $CO_2$. Below, we provide three independent lines of evidence, i.e., not using $LAI_{max}$ but atmospheric $CO_2$ measurements, to buttress our EC estimate.

**Independent lines of evidence.** First, the seasonal cycle of $CO_2$ concentration in the NHL, which shows a winter maximum due to respiratory processes and a late-summer minimum due to photosynthetic drawdown, may be considered as a proxy for NHL carbon exchange with the atmosphere[15–17]. Analyses of long-term measurements at NHL stations, Point Barrow (BRW, Alaska) and Alert Nunavut (ALT, Canada), reveal that this seasonal cycle has changed over time, dominated by a decreasing trend in the annual $CO_2$ minimum (Fig. 3a, b). Nearly all of this

change can be attributed to the land, as the trend in the abutting Arctic Ocean flux is ~15 times smaller (Fig. 3d; Methods). This strengthening of the seasonal swings of $CO_2$ concentration relates to photosynthesis rather than respiration changes[15–17] and thus features changes in GPP. So, if the EC estimate is closer to the true value of $\Delta$GPP at $2 \times CO_2$, then, models matching the EC estimate (e.g., MIROC-ESM) must also better simulate the changing $CO_2$ seasonal cycle measured at the NHL stations, in comparison to models that over- (e.g., HadGEM2-ES) or underestimate (e.g., CESM1-BGC). Indeed, the MIROC-ESM best reproduces the average observed seasonal cycle, and critically, the change in summertime minimum over time at both stations, in comparison to the other models (Fig. 3a, b). None of the models reproduce the observed phase of the seasonal cycle, which suggests a recurring problem among models in their representation of vegetation phenology[5]. Nevertheless, the model that projects $\Delta$GPP matching the $LAI_{max}$-based EC estimate is also the one

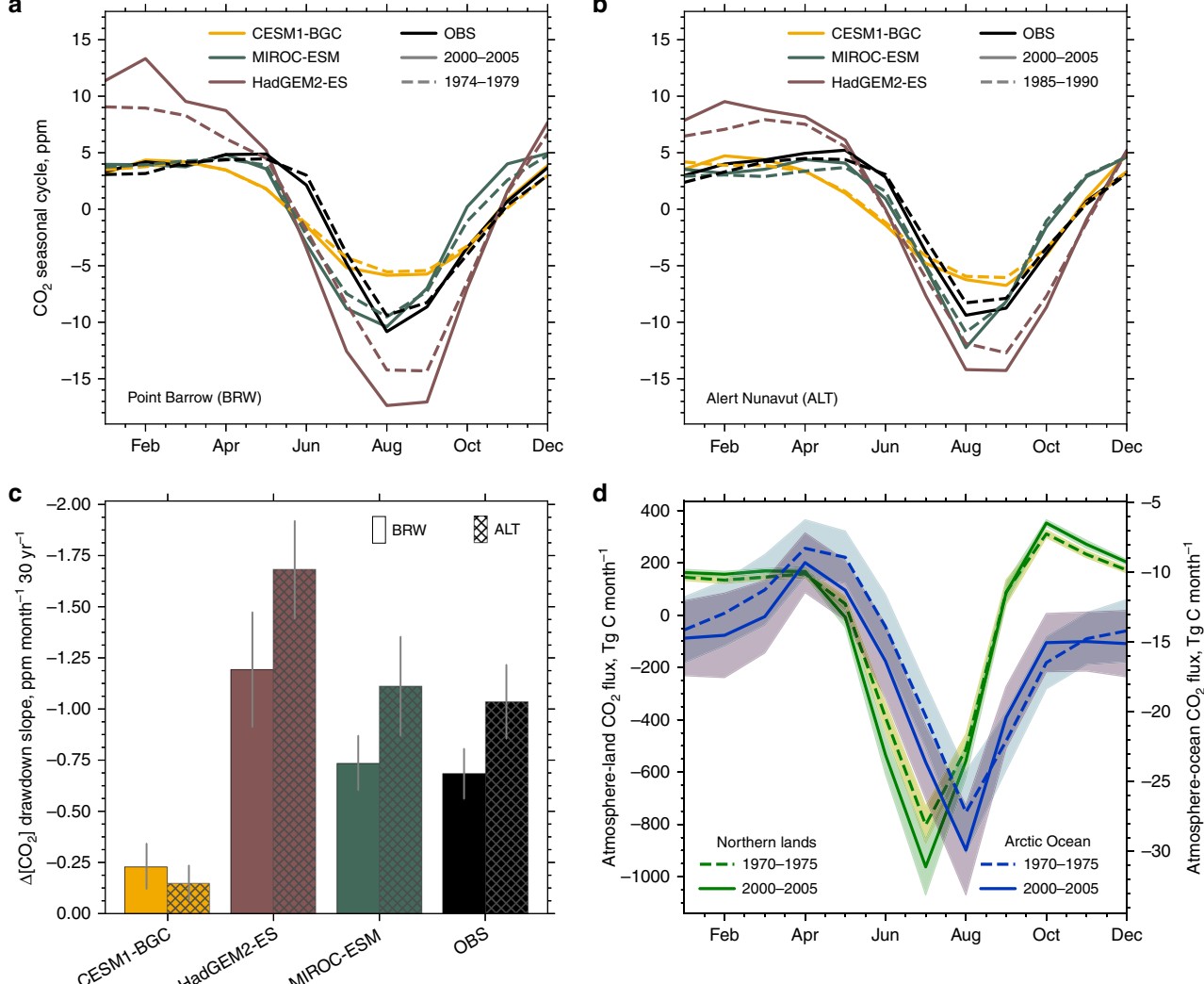

**Fig. 3** Lines of evidence in support of the Emergent Constraint estimate of NHL GPP. **a** Detrended seasonal cycle of Point Barrow (71.3°N, 203.4°E) $CO_2$ concentration at two time periods, 1974–1979 (dashed) and 2000–2005 (solid), from observations (black) and selected CMIP5 models (colored) spanning the full range of $LAI_{max}$ sensitivity (low-end: CESM1-BGC, closest-to-observations: MIROC-ESM, and high-end: HadGEM2-ES). **b** As in **a**, but showing the detrended seasonal cycle of Alert Nunavut (82.5°N, 297.7°E) $CO_2$ concentration at the two time periods, 1985–1990 (dashed) and 2000–2005 (solid). **c** Changes in the slope of summertime drawdown of $CO_2$ concentration over a 30-year period in representative models and observations at both stations (Methods). Gray bars denote one standard deviation. **d** Seasonal cycle of $CO_2$ fluxes into NHL land (green, 60°N–90°N, historical simulation, average of 3 realizations, MPI-ESM-LR) and Arctic Ocean (blue, ≥ 65°N, historical simulation, average of 10 realizations, MPI-ESM-HR), for two time periods, 1970–1975 (dashed) and 2000–2005 (solid). Shading indicates one standard deviation

that best captures the changes in observed seasonal cycle suggesting that the EC estimate, rather than the corresponding multi-model mean, best represents the true value of ΔGPP at $2 \times CO_2$. Thus, the multi-model mean is a large underestimate.

Second, measured changes in the amplitude of $CO_2$ seasonal cycle can be regarded as a metric of changes in NHL GPP[15–17,23]. This is not necessarily the case in ESMs, because uncertainty in wintertime carbon release processes influences considerably the annual $CO_2$ maximum and hence the amplitude—variations unrelated to photosynthetic activity[17]. To better isolate the effect of photosynthetic carbon fixation in the seasonal $CO_2$ signal, we use the slope of summertime drawdown instead of its amplitude. With the observed lengthening of the growing season[30] and general enhancement of GPP, the $CO_2$ concentration is increasingly tugged downward relative to the steady increasing trend. At both stations, the drawdown slope decreased over a period of 30 years (ALT: $-1.04 \pm 0.18$ ppm month$^{-1}$ 30 yr$^{-1}$ and BRW: $-0.68 \pm 0.12$ ppm month$^{-1}$ 30 yr$^{-1}$; Fig. 3c; Methods). The models also show a decreasing slope but disagree on the magnitude (Fig. 3c). Again, we note that the MIROC-ESM best reproduces the observed change in drawdown slope at both stations. Likewise, HadGEM2-ES considerably overestimates and CESM1-BGC underestimates the decline of the drawdown slope. According to the hypothesized EC approach (Supplementary Fig. 1), this is rooted in MIROC-ESM correctly capturing the sensitivity of an observable ($LAI_{max}$ in Fig. 2b or BRW and ALT drawdown slope in Fig. 3c) to $CO_2$ concentration. Consequently, this agreement in changes in $CO_2$ drawdown slope between long-term measurements and the closest-to-observations model in terms of greening sensitivity provides further support for the EC estimate of ΔGPP at $2 \times CO_2$ and suggests that the multi-model mean is a large underestimate.

Third, the available longest records of carbon exchange between the land/ocean and atmosphere (1980–2015) indicate that NHL lands changed from being a small carbon source in the early 1980s to a strong sink in the mid-2010s (Supplementary Fig. 5) meaning that the net biome production (NBP) increased— Jena CarboScope[32] (JENA) ΔNBP: $0.31 \pm 0.09$ Pg C yr$^{-1}$ and the Copernicus Atmosphere Monitoring Service[33] (CAMS) ΔNBP: $0.78 \pm 0.04$ Pg C yr$^{-1}$. NBP fluxes include emissions from disturbances, such as fire, and heterotrophic respiration, which may have increased due to warming over the period of record. Accordingly, the derived changes in NBP from the $CO_2$ inversion products can be considered as conservative estimates of NPP enhancement. The EC estimate using greening observations translates to a land net primary production (NPP) enhancement of about $0.32 \pm 0.02$ Pg C yr$^{-1}$, when adjusted for $CO_2$ concentration increase over the period of the atmospheric $CO_2$ inversion datasets (Methods). This estimate better agrees with the JENA estimate than the multi-model mean ($0.19 \pm 0.18$ Pg C yr$^{-1}$). All three, however, do not overlap with the CAMS estimate. Hence, the available evidence from inversion studies of atmospheric $CO_2$ measurements indicates NPP changes in NHL comparable to or larger than our EC estimate, and therefore the multi-model mean to be an underestimate.

## Discussion

The causes for model underestimation can perhaps be traced to the representation of carbon–nitrogen interactions and vegetation dynamics. Models that strongly underestimate (CESM1-BGC, NorESM-ME, and CanESM2) show excessive nitrogen limitation (in CanESM2, the $CO_2$ fertilization effect is down-regulated based on ambient and elevated $CO_2$ experiments)[9]. These models also lack the simulation of vegetation cover dynamics, and thus, do not reproduce the observed northward shift of vascular plants

and the associated higher productivity of shrubs and trees[6]. On the other hand, models that overestimate (HadGEM2-ES) show overly strong $CO_2$ fertilization effect and consequently excessive greening, presumably due to a lack of nitrogen limitation[23,34]. The model MIROC-ESM, which is closest to the EC estimate, stands out in its implementation of photosynthetic response to $CO_2$. Unlike the biogeochemical approach in other models, MIROC-ESM uses an empirical approach that implicitly includes nutrient limitation[6,35].

Although the Arctic represents only a small fraction of the terrestrial biosphere, the rapid climatic changes in NHL and uncertainties associated with the net carbon balance emphasize the need for further detailed analysis. The tendency for GPP underestimation in NHL by models reported here is also seen at the global scale (Supplementary Fig. 6). This, together with another recent study[23], suggests that most models are underestimating photosynthetic carbon fixation by plants and thus possibly overestimating atmospheric $CO_2$ and ensuing climatic changes[2,4,6].

## Methods

**Observational LAI product (LAI3gV1).** The new version (V1) of the LAI data set is an update of the widely used LAI3g data set[26]. It was generated using an artificial neural network (ANN) and the latest version (third generation) of the Global Inventory Modeling and Mapping Studies group (GIMMS) Advanced Very High Resolution Radiometer (AVHRR) normalized difference vegetation index (NDVI) data (NDVI3g). The latter has been corrected for sensor degradation, intersensor differences, cloud cover, solar zenith angle, viewing angle effects due to satellite drift, Rayleigh scattering, and stratospheric volcanic aerosols[36]. The ANN model was trained with overlapping data of NDVI3g and Collection 6 Terra MODIS LAI product[37,38], and then applied to the full NDVI3g time series to generate the LAI3gV1 data set. This data set provides global and year-round LAI observations at 15-day (bi-monthly) temporal resolution and 1/12 degree spatial resolution from July 1981 to December 2016. Currently, it is the only data set that spans this long period.

The quality of the previous version (V0) of the GIMMS LAI3g data set was evaluated through direct comparisons with ground-based measurements of LAI, indirectly with other estimates from similar satellite-data products, and also through statistical analysis with climatic variables, such as temperature and precipitation variability[26]. The LAI3gV0 data set (and related fraction vegetation-absorbed photosynthetically active radiation data set) has been widely used in various studies[5,15,19,20,31,39–41]. The new version LAI3gV1 used in our study is an update of that earlier version.

For both, observational and CMIP5 data, LAI is defined as the one-sided green leaf area per unit ground area in broadleaf canopies and as one-half the green needle surface area in needleleaf canopies. It is expressed in units of m$^2$ green leaf area per m$^2$ ground area. In this study, we use the annual maximum value of LAI, $LAI_{max}$, to quantify the greenness level of a surface. $LAI_{max}$ is less influenced by cloudiness and noise; accordingly, it is most useful in investigations of long-term greening and browning trends. The drawback of $LAI_{max}$ is the saturation effect at high LAI values[42]. However, this is less of a problem in high latitudinal ecosystems which are mostly sparsely vegetated, with $LAI_{max}$ values typically in the range of 2–3.

The bi-monthly GIMMS LAI3gV1 data are merged to a monthly temporal resolution by averaging the two composites in the same month. Then, for model and observational data alike, the two-dimensional global fields are cropped to the northern high latitudinal band defined as 60°N to 90°N, averaged in space and temporally reduced to the annual maximum value.

Although the AVHRR data underlying the LAI data in this study have corrections for various deleterious effects[36], the data may still contain residual non-vegetation-related effects. Therefore, we sought confirmation of the greening trend[19], on which the current study relies, from a more reliable but shorter record from the MODIS sensors[37,38]. These data are well calibrated, cloud-screened, and corrected for atmospheric effects, especially tropospheric aerosols. The sensor-platforms are regularly adjusted to maintain precise orbits. All algorithms, including the LAI algorithm, are physics-based, well-tested and currently producing the sixth generation data sets. The results, not shown here for brevity, illustrate global scale greening, across all latitudinal zones and broad vegetation classes. Zhang et al.[43] also reported matching greening trends between the latest (Version 6) MODIS and AVHRR (Version 3) vegetation index data sets.

We also found that the $LAI_{max}$ sensitivity derived with MODIS LAI data matched well with that obtained from the AVHRR LAI data (results not shown for brevity). Whether this indicates that the 17-year MODIS record from the period 2000 to 2016 captures information similar to the longer AVHRR record (1981–2016), or is simply a fortuitous occurrence, is not known, and deserves

further study. In the present context, however, this adds confidence to the AVHRR LAI data used in our study.

**Temperature data from ECMWF ERA-Interim.** Estimates of surface air temperature at 2 m height are from the widely used global atmospheric reanalysis product ERA-Interim by ECMWF[44] (for details see https://www.ecmwf.int/en/research/climate-reanalysis/era-interim). The global temperature fields were retrieved at a resolution of $0.5° \times 0.5°$ for monthly mean estimates derived from daily means. Other reanalysis products with similar specifications (NCEP/NCAR reanalysis, University of Delaware Air Temperature & Precipitation, and GHCN/CAMS reanalysis product) were also investigated. The differences among the various products were found to be minor.

**CMIP5 models used in this study.** In this study, we analyze a set of the most recent climate-carbon simulations of seven ESMs participating in the fifth phase of the Coupled Model Intercomparison Project, CMIP5[28]. The model data were obtained from the Earth System Grid Federation, ESGF (https://esgf-data.dkrz.de/projects/esgf-dkrz/). Seven ESMs provided output for the variables of interest for simulations esmHistorical, 1pctCO2, esmFixClim, and esmFdbk.

The esmHistorical simulation spans the period 1850–2005 and was driven by observed conditions such as solar forcing, emissions or concentrations of short-lived species and natural and anthropogenic aerosols or their precursors, land use, anthropogenic as well as volcanic influences on atmospheric composition. The models are forced by prescribed anthropogenic $CO_2$ emissions, rather than atmospheric $CO_2$ concentrations.

1pctCO2 is an idealized fully coupled carbon/climate simulation initialized from steady state of the pre-industrial control run and atmospheric $CO_2$ concentration prescribed to increase 1% $yr^{-1}$ until quadrupling of the pre-industrial level. The simulations esmFixClim and esmFdbk and are set up as the 1pctCO2 with the difference, that in esmFixClim (esmFdbk) only the radiative effect from increasing $CO_2$ concentration is included, while the carbon cycle sees the pre-industrial $CO_2$ level (vice versa)[28,45].

**Historical simulation with MPI-ESM higher-resolution setup.** MPI-ESM-HR is the coupled high-resolution setup of the latest version of the Max-Planck-Institute Earth System Model MPI-ESM1.2, which is the baseline for the upcoming Coupled Model Intercomparison Project Phase 6 (CMIP6). Here, the atmospheric component ECHAM6.3 has 95 vertical levels and twice the horizontal resolution (~100 km) than the CMIP5 version. The ocean component MPIOM is set up on a tripolar grid at nominal 0.4° horizontal resolution (TP04) and 40 vertical levels. MPI-ESM1.2 includes the latest versions of the land and ocean carbon cycle modules, comprising the ocean biogeochemistry model HAMOCC and the land surface scheme JSBACH. The forcing components for the historical simulation are chosen from CMIP5 (Methods) as at the time the simulations were conducted CMIP6 forcing was not available[46].

**Atmospheric $CO_2$ concentration data.** Monthly means of atmospheric $CO_2$ concentration at Point Barrow (71.3°N, 203.4°E) and Alert Nunavut (82.5°N, 297.7°E) are taken from the Global Monitoring Division measurement datasets (co2_brw_surface-insitu_1_ccgg_MonthlyData respectively co2_alt_surface-flask_1_ccgg_month) provided by the National Oceanic and Atmospheric Administration/Earth System Research Laboratory (NOAA/ESRL). Global monthly means of atmospheric $CO_2$ concentration are taken from the GLOBALVIEW-CO2 product (obspack_co2_1_GLOBALVIEWplus_v2.1_2016_09_02; for details see https://doi.org/10.15138/G3259Z) also available at NOAA/ESRL.

**Atmospheric $CO_2$ inversion products.** Atmospheric $CO_2$ inversions estimate surface–atmosphere net carbon exchange fluxes by utilizing $CO_2$ concentration measurements, a transport model and prior information on anthropogenic carbon emissions as well as carbon exchange between atmosphere and land respectively ocean[47]. We choose two products, which cover the longest time period (1980–2015) and are regularly updated, the Jena CarboScope[32] (JENA, version s81_v3.8, for details see http://www.bgc-jena.mpg.de/CarboScope/s/s81_v3.8.html) and the Copernicus Atmosphere Monitoring Service[33] (CAMS, version v15r2, for details see http://atmosphere.copernicus.eu/documentation-supplementary-products#greengas-fluxes) inversion systems. Both products provide monthly mean net flux estimates on a spatial resolution of 3.75° latitude and 5° longitude (JENA) and 1.875° latitude and 3.75° longitude (CAMS).

**Calculation of growing degree days above 0 °C (GDD0).** The global temperature fields from the reanalysis and model data are cropped to the northern high latitudinal band and averaged in space. The resulting one-dimensional time-series is converted to GDD above 0 °C by multiplying the days of each month with the respective monthly mean estimate if it is above 0 °C. Thus, we not only capture the warming signal, but also the prolongation of the growing season.

**Dimension reduction using principal component analysis.** The drivers GDD0 and atmospheric $CO_2$ concentration vary co-linearly due to the radiative effect of

increasing $CO_2$ concentration in the NHL. Thus, it is problematic to conduct an accurate factor separation in terms of their respective contribution to increase in $LAI_{max}$. However, the co-linearity suggests that a large amount of the signal is shared. Therefore, we conduct a PCA to apply dimension reduction[48].

The aim of the PCA is to find a linear combination of the driver variables that represents the one-dimensional projection with the largest possible variance. First, each driver time series $x_i$ is normalized by centering on its mean (subtracting $\bar{x}_i$) and scaling to unit variance (divide by one standard deviation $\sigma_i$). Thus,

$$\mathbf{X} = x_i' = \frac{x_i - \bar{x}_i}{\sigma_i}. \tag{1}$$

The matrix $\mathbf{X}$ contains the scaled time series $x_i'$ as columns. Next, we compute the covariance matrix $\mathbf{C}^X$,

$$\mathbf{C}^X = \frac{1}{n}\mathbf{X}^T\mathbf{X} \tag{2}$$

where $n$ is the length of each time series. The eigenvector $\mathbf{u}_k$ is obtained by solving the eigenvalue problem,

$$\mathbf{C}^X\mathbf{u}_k = \lambda_k\mathbf{u}_k. \tag{3}$$

The eigenvectors $\mathbf{u}_k$ are sorted according to the ordering of their associated eigenvalues $\lambda_k$. Projecting the initial driver matrix $\mathbf{X}$ onto the eigenvector $\mathbf{u}_1$ with the highest associated eigenvalue we arrive at the one-dimensional vector, the first principal component (PC),

$$\boldsymbol{\omega} = (\mathbf{X}\mathbf{u}_1)^T. \tag{4}$$

Transposed to a row vector, $\boldsymbol{\omega}$ denotes the time-series of the first PC, which explains the maximum variance of the two driver time series, atmospheric $CO_2$ concentration and GDD0.

**Estimation of historical $LAI_{max}$ sensitivity.** We derive the historical $LAI_{max}$ sensitivity applying a standard linear regression model ($f_n$)

$$f_n = a + bx_n \tag{5}$$

where $x_n$ denotes the driver time series, $a$ the intercept and $b$ the gradient. We obtain the best-fit line by minimizing the squared error ($s^2$)

$$s^2 = \frac{1}{N-2}\sum_{n=1}^{N}(y_n - f_n)^2 \tag{6}$$

where $y_n$ is the predictand time series and $N$ is the number of data points of each time series. The resulting best-fit gradient $b'$ represents the sought sensitivity. The standard error of $b$ and $a$ are given by

$$\sigma_b = \frac{s}{\sigma_x\sqrt{N}} \tag{7}$$

and

$$\sigma_a = s\sqrt{\frac{1}{N} + \frac{\bar{x}^2}{\sigma_x^2 N}} \tag{8}$$

respectively, for $\sigma_x$ being the standard deviation and $\bar{x}$ being the mean value of $x_n$.

**Derivation of changes in NHL $CO_2$ drawdown slope.** Graven et al.[17] showed that NHL $CO_2$ drawdown mostly happens in June and July. ESMs, however, disagree on the phase, mainly due to a premature start of the growing season (Fig. 3a, b). As a consequence, the $CO_2$ drawdown in models peaks earlier in the season. To obtain comparability for changes in $CO_2$ drawdown strength, we calculate the first derivative of the $CO_2$ concentration time series for the observational sites and each model individually. The annual minimum of the derivative in each time series reflects the months where the increase in photosynthetic $CO_2$ fixation is strongest ($CO_2$ drawdown slope). This procedure does not require a detrending of the atmospheric $CO_2$ signal.

For the BRW record, the 30 years of continuous overlap with the CMIP5 historical simulations were used to calculate the drawdown slopes (1974–2005). Due to the shorter overlap in the ALT record, 30 years of data from 1985 (start of measuring campaign) to 2015 were used for comparison with models. This is legitimate, because the $CO_2$ concentration rate of increment for both periods are just about the same. Model time series are obtained from the near-surface $CO_2$ concentration using the grid box in close proximity to each site. All yearly time series are slightly smoothed with a 2-year moving window. Changes are calculated from 5-year averages at the beginning and end of the record. Here, we only present a low-end, high-end and the closest-to-observation model from the greening EC

analysis, because Wenzel et al.[23] already reported the behavior of the entire CMIP5 ensemble in terms of simulating the NHL $CO_2$ seasonal cycle.

**Scaling of NPP estimates**. We scale and convert the EC estimate for changes in the GPP flux $\Delta F_{GPP,EC}$ for a doubling of the pre-industrial $CO_2$ level ($[CO_2]_{pi}$) to a NPP flux ($\Delta F_{NPP,EC}$) to obtain a comparable estimate to the atmospheric $CO_2$ inversion datasets using

$$\Delta F_{NPP,EC} = b \frac{\Delta[CO_2]_{1980-2015}}{[CO_2]_{pi}} \times \Delta F_{GPP,EC} \qquad (9)$$

where $\Delta[CO_2]_{1980-2015}$ denotes the change in atmospheric $CO_2$ concentration over the observational period from 1980 to 2015 and $b$ the standard GPP to NPP conversion factor of 0.5 (assuming uncertainty of 10%)[49,50].

**Comparison of C fluxes between Arctic Ocean and NHL land**. We require the use of a fully coupled ESM to separate between land and ocean in terms of the sign, magnitude, and seasonal cycle of the respective net carbon exchange fluxes with the atmosphere. We have access to a spatially-high resolved historical run (10 realizations) of the MPI-ESM which has the ability to reproduce seasonality in the Arctic Ocean (Methods). The terrestrial carbon pools have not been brought into equilibrium due to computational limitations in these high-resolution simulations. Thus, we use simulations from the same model but at low spatial resolution (3 realizations), the CMIP5 esmHistorical simulation, to address land carbon exchange fluxes (Methods).

The NHL land sink is approximately 2.5 times larger than the Arctic Ocean sink, on an annual basis. However, in terms of the change in carbon sink between the mid-1970s and early-2000s, the increase in $CO_2$ uptake by the land is about 15 times larger than the ocean. Accordingly, the Arctic Ocean can be ignored when trying to explain changes during the recent past, i.e., BRW period of $CO_2$ concentration measurements.

During the months from May to September (may-to-sep) when photosynthetic $CO_2$ drawdown is happening, the change in land sink is about 0.4 Pg C on an annual basis. Especially between May and July, the $CO_2$ concentration is rapidly declining, i.e., photosynthesis prevails $CO_2$ release processes. Thus, nearly the entire increase of 0.4 Pg C can be attributed to increasing NPP. The EC analysis shows that the MPI-ESM model is rather close to observations but generally underestimating greening sensitivity and thus also the GPP enhancement. These results are not provided as further proof of the EC estimate, although they are not contradictory, they are provided to compare the strength of NHL land and Arctic Ocean carbon sinks and why the ocean component can be neglected.

**Bootstrapping for probability estimation**. We apply bootstrapping to estimate the 68% confidence of the emergent linear relationship due to the small sample size of the CMIP5 ensemble. First, we randomly resample the data with replacement, where the size of the resample is equal to the size of the original sample $N$. Second, we compute the least-squares linear best fit for the resampled data. Third, we repeat this procedure $m$ times (minimum $m = 100$) until the difference between the median best fit line of $m - 1$ and $m$ computed regressions approaches zero (the actual threshold was set to a difference less than 1%). We derive the 68% confidence contours of equal probability based on the set of $m$ random regression lines.

**Calculation of probability density functions**. We derive a probability density function (PDF) for the observed sensitivity $b'$ (associated standard error $\sigma_b$, Methods)

$$P(b) = \frac{1}{\sqrt{2\pi\sigma_b^2}} \exp\left\{-\frac{(b-b')^2}{2\sigma_b^2}\right\} \qquad (10)$$

assuming Gaussian distribution. The PDF of $y$ for given $x$,

$$P(y|x) = \frac{1}{\sqrt{2\pi\sigma_f^2}} \exp\left\{-\frac{(y-f(x))^2}{2\sigma_f^2}\right\}, \qquad (11)$$

represent the contours of equal probability density around the best-fit linear regression, where $\sigma_f$ denotes the 68% confidence contours estimated by boot-strapping (Methods). As shown in Cox et al.[21], for a given observation-based PDF $P(b)$ and a model-based PDF $P(y|x)$, the PDF of the EC on $y$ is

$$P(y) = \int_{-\infty}^{\infty} P\{x|y\}P(x)\,dx. \qquad (12)$$

The PDF of the CMIP5 unweighted multi-model mean is configured assuming Gaussian distribution.

**Code availability**. The code used in this study is available from the corresponding author upon request.

## Data availability

All data used in this study are available from public databases or literature, which can be found with the references provided in respective Methods section. Processed data is available from the corresponding author upon request.

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

## Acknowledgements

The authors acknowledge T. Park and C. Chen for their help with remote sensing data. The authors thank W. Müller, H. Li, and T. Ilyina for providing the high-resolution MPI-ESM simulations. Further, the authors thank H. Graven for assistance with the Point Barrow $CO_2$ data, R. Keeling for advice on the seasonal cycle of $CO_2$, M. Heimann, C. Rödenbeck, and P. Cias for advice on atmospheric $CO_2$ inversion, and G. Lasslop for reviewing the manuscript. R.B.M. acknowledges funding by the NASA Earth Science Directorate and the Alexander von Humboldt Foundation. This work contributes to the H2020 project CRESCENDO, which receives funding from the European Union's Horizon 2020 research and innovation program under Grant Agreement No. 641816. The article processing charges for this publication were paid by the Max Planck Society.

## Author contributions

A.J.W. performed the research. All authors contributed ideas and to the interpretation of the results. R.B.M. and A.J.W. drafted the manuscript with inputs from G.A.A. and V.B.

## Additional information

**Competing interests:** The authors declare no competing interests.

