## [Peer Review File · Nature Communications]

Reviewers' comments:

Reviewer #1 (Remarks to the Author):

Winkler et al. apply an Emerging Constraint (EC) approach by combining recent climate and satellite LAI observations to estimate GPP sensitivity to forcing. This sensitivity is then extrapolated to the future to conclude that CMIP5 ESMS underestimate the land's ability to assimilate atmospheric CO₂. The EC approach is interesting, but I agree with the previous reviewers that its application must be more carefully considered. I think the authors have not sufficiently demonstrated that they have met the basic assumptions underlying the EC approach, for several reasons (some of which were raised by previous reviewers):

1. The Emerging Constraint (EC) approach requires several assumptions be satisfied, e.g.:

a. The models used for extrapolation must believably represent the majority of processes responsible for system dynamics over the current and extrapolated time frames.

i. The CMIP5 models used in this study have been analyzed in many studies, and they have been shown to either poorly represent, or not even include, the dominant processes important for high-latitude hydrological, geomorphological, coupled carbon and nutrient, thermal, or species distribution changes (Bouskill et al., 2014; Todd-Brown et al., 2013; He et al., 2016; Koven et al., 2013; De Kauwe et al., 2017; Zaehle et al., 2014). For example, given that high-latitude systems (1) are often strongly N limited; (2) CO₂ fertilization requires additional N supply; and (3) none of the tested models even represent nutrient constraints, it is hard to believe they are even predicting the current constraints on GPP reasonably, much less the future responses. Given this, and other, evidence, I do not find it believable that the CMIP5 simulations can be used to infer high-latitude GPP sensitivity to climate over decades.

b. That extrapolation must be linear with respect to the independent variable being used (i.e.,).

i. Although the models show this linearity, there is no demonstration that the real world follows a linear relationship. Current understanding of high-latitude system dynamics indicates that these ecosystems will respond non-linearly to potentially dramatic changes in geomorphology, hydrology, plant distributions, fire, and nutrient availability. Given that the CMIP5 models clearly have unrealistic representations of high-latitude processes, using them to make an argument about linearity seems inappropriate.

2. Given: (1) all the complexity of linking changes in NDVI to changes in LAI to changes in GPP, especially for high latitude systems; (2) no assessment of the uncertainties in this calculation; and (3) no demonstration using any ground observations to test whether the hypothesized links between GPP and are accurate, it does not seem reasonable to conclude that the calculated GPP versus relationship is sufficiently robust to justify an extrapolation to the future.

Bouskill, N. J., Riley, W. J., and Tang, J. Y.: Meta-analysis of high-latitude nitrogen-addition and warming studies implies ecological mechanisms overlooked by land models, *Biogeosciences*, 11, 6969-6983, doi:10.5194/bg-11-6969-2014, 2014.

De Kauwe, M. G., Medlyn, B. E., Walker, A. P., Zaehle, S., Asao, S., Guenet, B., Harper, A. B., Hickler, T., Jain, A. K., Luo, Y. Q., Lu, X. J., Luus, K., Parton, W. J., Shu, S. J., Wang, Y. P., Werner, C., Xia, J. Y., Pendall, E., Morgan, J. A., Ryan, E. M., Carrillo, Y., Dijkstra, F. A., Zelikova, T. J., and Norby, R. J.: Challenging terrestrial biosphere models with data from the long-term multifactor Prairie Heating and CO₂ Enrichment experiment, *Global Change Biology*, 23, 3623-3645, 2017.

He, Y. J., Trumbore, S. E., Torn, M. S., Harden, J. W., Vaughn, L. J. S., Allison, S. D., and

Randerson, J. T.: Radiocarbon constraints imply reduced carbon uptake by soils during the 21st century, *Science*, 353, 1419-1424, 2016.

Koven, C. D., Riley, W. J., and Stern, A.: Analysis of Permafrost Thermal Dynamics and Response to Climate Change in the CMIP5 Earth System Models, *J Climate*, 26, 1877-1900, Doi 10.1175/Jcli-D-12-00228.1, 2013.

Todd-Brown, K. E. O., Randerson, J. T., Post, W. M., Hoffman, F. M., Tarnocai, C., Schuur, E. A. G., and Allison, S. D.: Causes of variation in soil carbon simulations from CMIP5 Earth system models and comparison with observations, *Biogeosciences*, 10, 1717-1736, Doi 10.5194/Bg-10-1717-2013, 2013.

Zaehle, S., Medlyn, B. E., De Kauwe, M. G., Walker, A. P., Dietze, M. C., Hickler, T., Luo, Y. Q., Wang, Y. P., El-Masri, B., Thornton, P., Jain, A., Wang, S. S., Warlind, D., Weng, E. S., Parton, W., Iversen, C. M., Gallet-Budynnek, A., McCarthy, H., Finzi, A. C., Hanson, P. J., Prentice, I. C., Oren, R., and Norby, R. J.: Evaluation of 11 terrestrial carbon-nitrogen cycle models against observations from two temperate Free-Air CO₂ Enrichment studies, *New Phytologist*, 202, 803-822, 2014.

Reviewer #2 (Remarks to the Author):

Review of "Earth System Models Underestimate Carbon Fixation by Plants in the High Latitudes" by Alexander Winkler et al.

Reviewed by Matthias Forkel, 2018-02-06

This manuscript uses the emergent constraints concept to estimate the change in northern-hemisphere land GPP in response to a doubling of CO₂ in comparison to pre-industrial conditions. The study has a strong similarity to a previous study by Wenzel et al. (2016). The main difference is that this study uses satellite observations of leaf area index to derive the emerging constraint on the GPP change. This is a novel approach and in my view a major advance in comparison to Wenzel et al. (2016). The study further provides three other evidences for GPP change that are independent of the LAI data in order to support their main estimate. The study concludes that Earth system models underestimate the change in carbon uptake in response to CO₂ change. I was asked to comment on the revisions and whether the reviewers' concerns were adequately addressed. In my view, the responses to the reviewers were adequate. However, some of the responses are only partly reflected in the revised manuscript. I'm convinced that the manuscript is suited for publication in *Nature Communications* but I think the following changes are necessary:

- Change the title and adapt the scope of the study from "Models Underestimate ..." to the presentation of a new EC-approach to estimate $\Delta\text{GPP}/2 \cdot \text{CO}_2$ (see Comment 1).
- Include also a short discussion of reasons (processes) that contribute to the fact that some models over-/underestimate the changes (Comment 1).
- Remove the estimation of NPP changes (evidence approach 1). This approach raises more questions and does not provide a reliable evidence for the LAI-based estimates (Comment 2)
- Include a comparison of recent $\Delta\text{GPP}/2 \cdot \text{CO}_2$ estimates and replace Figure 3 with a new Figure (Comment 3).
- Clearly define and consistently use your terms. It is often not clear if you mean the sensitivity of LAI_{max} to CO₂+GDD0 or the sensitivity of ΔGPP to LAI, or if you mean GPP, NPP, or the overall land carbon uptake (NBP) (see my Specific Comments).

--- Comment 1: Conclusion, title and scope of the study

The main conclusion (= title) of the study is "Earth System Models Underestimate Carbon Fixation by Plants in the High Latitudes". However, only seven models are used, one model (HadGEM2-ES) overestimates ΔGPP , one models are likely within the uncertainty of the observational-derived

Delta-GPP (MIROC-ESM, MPI-ESM-LR), and only four models underestimate Delta-GPP (Fig. 2c). Also the probability density functions of the Delta-GPP distribution from the emerging constrain and from the multi-model mean largely overlap. So the question is, if this difference is really significant ($n = 7$ models!) to conclude that "Earth System Models Underestimate Carbon Fixation by Plants in the High Latitudes". For sure, you can conclude that some models underestimate. However, it does not matter if ESMs – in average – under- or overestimate land carbon uptake. The results nicely describe the variety of LAI, LAI changes, LAI-PC1 sensitivities in the models but then the remaining text focusses on conclusions about the multi-model mean, which has likely a very limited meaning if the models show such a variety. What really matters is how individual models perform and which processes contribute to an over- or underestimation. In order to provide a new insight, the text (and title) should either focus on 1) the novel approach to use LAI data as an emergent constraint for GPP changes or 2) an analysis and discussion about land-atmosphere processes and feedbacks that affect GPP change in the observational world and in the world of each model. Such a discussion should cover potential impacts on GPP through changes in seasonality/phenology, nutrient availability, disturbances and vegetation dynamics (increase in tree and shrub cover), permafrost and water availability (reviewer 2).

--- Comment 2: Derivation and scaling of NPP estimates

The author present an estimate of NPP change in the NHL latitudes as an independent estimate or evidence for their findings. To estimate NPP, the annual minimum of the seasonal cycle of CO₂ is converted to a change in land carbon flux (chapter 5 in Methods). Reviewers 1 and expressed their concern about this approach.

I'm not convinced by this approach. First, this approach is presented without any references. The authors provide no evidence that the approach is valid. The used terminology is also not well defined. Delta-F_C, BRW is estimated from CO₂ minimum. Delta-F_NPP, EC is estimated from Delta-F_GPP, EC. Which metrics are then used and compared in the main text (P5 L10-22)?

Second, the approach is based on several assumptions which are definitely not true: 1) the change in the CO₂ minimum is only affected by NPP, 2) a constant conversion factor of CO₂ fractions (ppm) to carbon flux (PgC), and 3) a constant contribution of high latitude areas to the CO₂ minimum. Various studies investigated the influencing factors on the seasonality of CO₂ (Randerson et al., 1997; Graven et al., 2013; Forkel et al., 2016). From these studies, we know that

- 1) changes in CO₂ seasonality are affected by productivity and respiration (less by other carbon fluxes),
- 2) that the contribution of GPP and ecosystem respiration to the CO₂ amplitude changes over time (e.g. Tab S6 in (Forkel et al., 2016)), and
- 3) the contribution of different regions (e.g. Arctic or boreal region) also changed over time depending on the magnitude of the regional changes in land carbon uptake. For example, arctic regions contribute around 26% and 32% and boreal regions 54 and 52% to the seasonal amplitude of CO₂ at BRW in the 1970s and 2000s, respectively (Tab. S5 in (Forkel et al., 2016)).

In summary, this approach provides more questions than it provides a robust and reliable independent estimate of the emerging constraint. The approach does not at all support the findings of the study but has for me the potential to destroy the entire flow of the paper. Please remove these estimates from the paper.

However, it is a good idea and likely possible to use the seasonality of CO₂ as constraint for GPP. For example, a relationship between changes in GPP and changes in CO₂ amplitude is presented in Figure 3 in Forkel et al. (2016).

Furthermore GPP products that were upscaled from eddy covariance observations (Tramontana et al., 2016) can be used to derive the Delta-GPP/LAI_max sensitivities from more reliable data-driven estimates. Such a dataset also has limitations. For example, it is unclear how well it reproduces the CO₂ fertilization effect and of course it also does not "see" a period with 2*CO₂ increase. However, I'm still convinced that such an approach is more reliable than estimating Delta-NPP from the CO₂ minimum.

--- Comment 3: Include MODIS-based EC estimates and compare multiple EC estimates

Reviewer 1 mentioned that the results could be affected by the uncertainty in satellite estimates of LAI. When I was reading the manuscript (before reading the reviewer comments and responses), I had the same question. The authors provided a good response to the reviewer comment. However, this is not appropriately reflected in the revised manuscript. Figure R1.2 should be included in the manuscript (Extended Data). Moreover, the MODIS-based estimates should be also included Fig. 2 to demonstrate the effect of data uncertainty in the derived EC estimates.

As the study provides four different estimates of Delta-GPP and it is an improved follow-up of the work by Wenzel et al. (2016), I advise you to show all recent estimates in one plot. The Delta-GPP/2*CO₂ estimates from LAI3g and MODIS, from the ESMs, from Wenzel et al. (2016), from the upscaling GPP, and the Delta-NBP/2*CO₂ estimates from the inversions should be all shown in one plot (barplot with error estimates). This figure will provide insight in the current uncertainty and the most likely values of Delta-GPP/2*CO₂ and will also help to compare your results with the results from Wenzel et al. (2016). Replace the current Figure 3 with this new figure.

--- Specific Comments

P2 L1-3: Please provide a reference for any numbers that are not yours.

P2 L3-6: I propose to split this sentence into two (e.g. after "regions") to improve readability.

P2 L9: Reference 13 (Forkel et al., 2014) does not fit to this sentence; throughout the paper this reference does not fit. Probably you intended to refer to my other paper: (Forkel et al., 2016). I also propose to place reference 13 together with references 14 and 15.

P3 L30: Can you refer to a figure where the linear relation can be seen? – Figure 2a? The sentence does not really fit as a topic sentence for this paragraph, but it rather describes a result and hence should come later. It would even better fit as a summary sentence of this paragraph. You should rather start the paragraph with an introductory sentence about the general sensitivity of LAI_{max} to CO₂ and temperature (or GDD).

P4 L5: "span a large range in a somewhat complex manner" – This expression can mean anything. I don't understand what you want to say here and how this links with Fig. 2 a and b. Please be more specific.

P4 L1-16: In order to make the text better understandable, I propose to define the term greening "rate" or "sensitivity" explicitly already at the beginning and then use one of the two terms consistently throughout the manuscript. However, I understand "rate" rather as a change in time. So I would use "greening rate" for temporal trends in LAI_{max}, "greening sensitivity" for the slope between LAI_{max} and PC1, and "LAI_{max}-GPP sensitivity" for slopes between Delta-LAI_{max} and Delta-GPP.

P4 L21-23: It is not clear in this sentence if you mean the sensitivity of LAI_{max} to PC1 (as suggested by the reference to Fig 2) or to Delta-GPP (as suggested by the references to Extended Data Figures 2 and 3 in the previous sentence). This can be solved if you define the terms already at the beginning (see previous comment).

P4 L25-27: "linear trend": Do you mean linear "relation"? I associate "trends" rather with temporal changes.

P5 L13: You can also add the updated reference 13.

P5 L14-16: You make a very big step in this sentence! Although you refer to Methods S5, it is crazy to just jump from CO₂ minimum changes to a change in NPP. For me as a reader, this just creates a feeling like "Are you joking? Now I don't trust you". You really should add here a sentence how you get to this NPP change.

P5 L17-18: NPP is a flux, i.e. it is per time (e.g. x PgC yr⁻¹). If one describes a change in NPP per time (e.g. per year) the unit is PgC yr⁻². I'm not certain how the "NPP enhancement" was calculated and what the 0.46 PgC yr⁻¹ really mean. Should the unit be PgC yr⁻²? If not is this then an absolute change between 1960s and 2010s?

P5 L19-20: I don't understand the sentence. NPP is independent of Rh. But NBP or NEE include Rh (Chapin et al., 2006; Schulze, 2006):

$$\text{NPP} = \text{GPP} - \text{Ra}$$

$$\text{NBP} = (\text{Rh} + \text{Ra} + \text{FireEmissions} + \text{other}) - \text{GPP}$$

$$\text{NEE} = (\text{Rh} + \text{Ra}) - \text{GPP}$$

If Rh increased, why should the NPP trend be larger? Changes in respiration contributed to a decrease of the CO₂ amplitude at BRW because respiration in northern latitudes happens more during summer because of higher temperatures. However, GPP increases contributed to a much larger increase in CO₂ amplitude than respiration caused a decrease, which overall results in an increasing CO₂ amplitude (Tab S6 in (Forkel et al., 2016)). These results likely also apply to the CO₂ minima because changes in the CO₂ amplitude are largely caused by changes in CO₂ minima: An increase in Rh contributes hence to an increase in CO₂ minima.

P5 L28: Again the updated reference 13 would better fit.

P6 L5-6: If you refer to results for the Alert site, show them! It does not make sense to make the reader curious about results and then "close the door" with "results not shown". One can see this impolite behaviour in many studies but it is completely useless. That means: If you refer to results, show them (i.e. include Fig R1.4). If you don't want to show them for brevity, don't mention them.

P6 L18-20: Again, this sentence does not fit. The inversions quantify the net land exchange (NBP) and do not provide information on the component fluxes NPP and Rh.

P6 L24-30: I'm missing a short discussion on what processes the models miss or which process are miss-represented in order to reproduce 1) the correct LAI changes, and 2) the GPP change.

P11 Chapter 1.1: This chapter can be shortened because it mostly describes previous published studies about LAI and LAI product generation. I propose to write only the essential information for your study and what you did with the data.

P12 L21-26: When I was reading the manuscript, I was wondering already if the results will be the same with MODIS data. Please don't tell me that you don't show it for brevity. I really want to see the EC estimate that is based on MODIS data. Include Figure R1.2 in the text and add the MODIS-based results to all figures of the main text.

P15 L26 – P16 L10. This is basically a linear regression. The text reads like Cox et al. would have invented linear regression. I think this chapter can be significantly shortened.

P18 L21 – P19 L19: Why you need to calculate PDFs? You could also just show the uncertainty in the EC estimate based on the distribution of the computed EC values.

Figure 1a: Please describe in the caption how LAI_max and PC1 were spatially integrated (averaged?).

Figure 2c: Most of the paper describes that the multi-model mean underestimates the emerging constraint. Therefore, I propose to show the multi-model mean in this figure to visualize the underestimation. Additionally, the uncertainty estimate (or standard error) of the observational-derived Delta-GPP estimate should be added as shaded area round the dashed line.

Figure 2d: I don't understand why the Delta-GPP estimate is shown as PDF. I would rather prefer to see estimates for individual models and multi-model mean. Please show the results as barplot like in Fig. 2 b because this shows just the data as it is and does not rely on the assumption of normal distribution. It would also help to read the manuscript because as a reader one is already used to the barplot. There is no need to mimic all the PDF-like figures from the Cox et al. and Wenzel et al. studies!

Figure 3: Actually, I don't understand how these datasets provide evidence for the "emerging constraint estimate of NHL GPP" and how they are connected with the previous figures. These datasets all show an increase of the land carbon uptake which is however known from previous studies working with these datasets (Graven et al., 2013; Forkel et al., 2016; Wenzel et al., 2016). But where is the link to the emerging constraint on Delta-GPP? Reviewer 3 also mentions that this figure repeats just previous findings. Please move this figure to the supplement and replace it with a figure (e.g. barplot) that provides an overview of recent Delta-GPP/2*CO₂ estimates (i.e. emerging constraints, inversion-based results, results based on CO₂ amplitude).

Extended Data Figure 2: Please indicate if this also integrates over the NHL regions.

Extended Data Figure 3: From the caption, I understand that blue is $[2 \times \text{CO}_2] - [1 \times \text{CO}_2]$ but from the legend blue is " $< 2 \times \text{CO}_2$ " which I don't understand. So I don't really understand what model outputs were used or selected to compute Delta_LAI and Delta_GPP for this figure. Please make the caption and legend consistent. Additionally, the dots and lines from Extended Data Figure 2 could be added to this plot which would reduce the number of figures.

Extended Data Figure 4: The figure is very difficult to understand because the colours in the legend do not fit with shown symbols and lines. For example, red squares show "CO₂ fertilization" effect but what are then the squares in the other colours? Additionally, what purpose has the legend for the models? Bold lines with colours for each model are not in this plot. And the green fitting line is a fit through all crosses, even if they are not green-coloured? You need to definitely improve this plot (fit legend symbol/colours with the data), otherwise one is only puzzled with linking the symbols and this distracts completely from the message of the figure.

Extended Data Figure 5: You always say that the multi-model mean underestimates the EC estimate. So can you add a cross for the multi-model mean to show this underestimation?

--- References

- Chapin, F.S., Woodwell, G.M., Randerson, J.T., Rastetter, E.B., Lovett, G.M., Baldocchi, D.D., Clark, D.A., Harmon, M.E., Schimel, D.S., Valentini, R., Wirth, C., Aber, J.D., Cole, J.J., Goulden, M.L., Harden, J.W., Heimann, M., Howarth, R.W., Matson, P.A., McGuire, A.D., Melillo, J.M., Mooney, H.A., Neff, J.C., Houghton, R.A., Pace, M.L., Ryan, M.G., Running, S.W., Sala, O.E., Schlesinger, W.H. & Schulze, E.-D. (2006) Reconciling Carbon-cycle Concepts, Terminology, and Methods. *Ecosystems*, 9, 1041–1050.
- Forkel, M., Carvalhais, N., Rödenbeck, C., Keeling, R., Heimann, M., Thonicke, K., Zaehle, S. & Reichstein, M. (2016) Enhanced seasonal CO₂ exchange caused by amplified plant productivity in northern ecosystems. *Science*, aac4971.
- Forkel, M., Carvalhais, N., Schaphoff, S., v. Bloh, W., Migliavacca, M., Thurner, M. & Thonicke, K. (2014) Identifying environmental controls on vegetation greenness phenology through model–data integration. *Biogeosciences*, 11, 7025–7050.
- Graven, H.D., Keeling, R.F., Piper, S.C., Patra, P.K., Stephens, B.B., Wofsy, S.C., Welp, L.R., Sweeney, C., Tans, P.P., Kelley, J.J., Daube, B.C., Kort, E.A., Santoni, G.W. & Bent, J.D. (2013) Enhanced Seasonal Exchange of CO₂ by Northern Ecosystems Since 1960. *Science*, 341, 1085–1089.
- Randerson, J.T., Thompson, M.V., Conway, T.J., Fung, I.Y. & Field, C.B. (1997) The contribution of terrestrial sources and sinks to trends in the seasonal cycle of atmospheric carbon dioxide. *Global Biogeochemical Cycles*, 11, 535–560.
- Schulze, E.-D. (2006) Biological control of the terrestrial carbon sink. *Biogeosciences*, 3, 147–166.
- Tramontana, G., Jung, M., Schwalm, C.R., Ichii, K., Camps-Valls, G., Ráduly, B., Reichstein, M., Arain, M.A., Cescatti, A., Kiely, G., Merbold, L., Serrano-Ortiz, P., Sickert, S., Wolf, S. & Papale, D. (2016) Predicting carbon dioxide and energy fluxes across global FLUXNET sites with regression algorithms. *Biogeosciences*, 13, 4291–4313.
- Wenzel, S., Cox, P.M., Eyring, V. & Friedlingstein, P. (2016) Projected land photosynthesis constrained by changes in the seasonal cycle of atmospheric CO₂. *Nature*, 538, 499–501.

Reviewer #3 (Remarks to the Author):

This is a carefully executed analysis which significantly adds to the literature using emergent constraints to get more reliable estimates of key quantities in the global carbon cycle. I am satisfied with the authors' in-depth responses to the previous review comments. For my part I have only a few points to add:

1. I still do not understand why there is a discrepancy between results of the EC approach and the

conclusion of Thomas et al. that *all* models (in the MsTMIP ensemble) underestimate the historical response of LUE to CO₂. The present MS seems to indicate that it is just a matter of a different set of models. But Graven et al.'s (2013) Science paper already included a graph showing the failure of all CMIP5 models to reproduce the percentage amplification of the NHL seasonal cycle.... It would be most helpful if the authors could shed some more light on this issue, and important that they should explain why their conclusion appears to differ from that of Graven et al. (2013).

2. The authors cite the flawed study of Kolby Smith and others apparently showing that models over-estimate the CO₂ fertilization effect. Either this reference should be deleted, or the subsequent correspondence item also in NCC debunking Smith et al. should also be cited. (I declare an interest as a co-author of the correspondence item, but I believe my point is valid nonetheless!)

3. It seems to me that the authors may have missed an opportunity to make separate estimates of the physiological and climate effects. They are not separable in terms of the overall trend, however the fact that climate shows strong interannual variability while CO₂ show a monotonically increasing trend should allow the different effects to be separated statistically.

Colin Prentice

Authors Response to Referee 1 (Nature Communications manuscript 17-32151-T)

Winkler et al. apply an Emerging Constraint (EC) approach by combining recent climate and satellite LAI observations to estimate GPP sensitivity to forcing. This sensitivity is then extrapolated to the future to conclude that CMIP5 ESMs underestimate the land's ability to assimilate atmospheric CO₂. The EC approach is interesting, but I agree with the previous reviewers that its application must be more carefully considered. I think the authors have not sufficiently demonstrated that they have met the basic assumptions underlying the EC approach, for several reasons (some of which were raised by previous reviewers):

- We thank the referee for a critical review of our research. However, we respectfully contest the statement that our work has not sufficiently met the basic assumptions underlying the EC approach.

1.1 The Emerging Constraint (EC) approach requires several assumptions be satisfied, e.g. the models used for extrapolation must believably represent the majority of processes responsible for system dynamics over the current and extrapolated time frames.

- The general consensus is that current generation ESMs represent most of land surface biophysical and biogeochemical processes, as evidenced by their extensive use, not only by the scientific community but also in policy prescribing activities (e.g. the IPCC). ESMs include physical interactions between the atmosphere, ocean, and land components, that are absent in the offline land surface models often used in model evaluation. Weaknesses in ESMs are acknowledged and being refined. Given that the models are “not perfect”, methods such as the EC are crucial to extract information which is more robust than outputs from individual models.

1.2 The CMIP5 models used in this study have been analyzed in many studies, and they have been shown to either poorly represent, or not even include, the dominant processes important for high-latitude hydrological, geomorphological, coupled carbon and nutrient, thermal, or species distribution changes (Bouskill et al., 2014; Todd-Brown et al., 2013; He et al., 2016; Koven et al., 2013; De Kauwe et al., 2017; Zaehle et al., 2014).

- CMIP5 models represent dominant physical and biogeochemical processes in the Arctic, the most important being atmospheric circulation, clouds, hydrological cycle, sea ice, vegetation biophysics and photosynthesis. We agree that the models are not perfect, which is the reason for model intercomparison projects such as CMIP5. The EC approach relies on the dynamics of ESMs in a multi-model ensemble and tries to overcome the limitations of each individual model. The papers cited by the referee are tangential/not-relevant to our research as discussed below:
 - Bouskill et al. (2014) analyze simulations of two derivatives of a single land model (CLM-Century, CLM-CN) in a stand-alone setup, using atmospheric forcing fields. It is questionable how their results are relevant to an evaluation of the coupled model CMIP5 ensemble used in our study. Also, the focus is belowground carbon dynamics under warming and nitrogen addition. Our study, however, is concerned with vegetation greening and the connection to photosynthesis, i.e. above-ground processes. In a short section about aboveground dynamics (page 6975, 4.1.2 Aboveground dynamics), Bouskill et al., (2014) refer to field studies that report increase of woody vegetation and a general stimulation of aboveground activity under warming, in line with CMIP5 models and the conclusions of our study.

- Todd-Brown et al. (2013) exclusively look into soil carbon stocks in CMIP5 and conduct a global comparison with observational datasets that disagree with each other as much as the ESMs. Our study is concerned with above-ground processes in the northern high latitudes – so this study of carbon stocks or soil carbon on a global scale is tangential to our work.
- The work of He et al., (2016) quoted by the referee actually provides a good example how one can use the dynamics of ESMs in a multi-model ensemble together with observations (soil radiocarbon measurements). Using CMIP5 simulations (1pctCO2, esmFixClim1) which have been also used in our study, the authors report an observational constraint on carbon sequestration potential of soils. We also note that this work is tangential to our research of above-ground processes.
- The study by Koven et al. (2012) is not relevant to our work. The authors present a comparison of CMIP5 models with observational datasets with respect to physical aspects of permafrost soils as well as provided projections under different scenarios. Although a wide inter-model spread in CMIP5 ensemble is reported, it is not clear how this is relevant to the specifics of our research.
- Zaehle et al. (2014) evaluate a set of terrestrial carbon-nitrogen models against observations of photosynthesis, biomass and soil carbon at two FACE experimental sites in the temperate climatic zone. The performance of the models is mixed, some aspects are represented well, others limited or not at all. They conclude that ecosystem models are able to capture important features of the experiments providing support to their projections. However, the response of terrestrial ecosystems to elevated CO₂ is still insufficiently understood and therefore the authors call for observation-based constraints, such as the study we report. Please also see our response to comment 1.3.
- De Kauwe et al. (2017) extends the analysis of Zaehle et al. (2014). They evaluate terrestrial biosphere models against a single multi-factor Prairie Heating and CO₂ Enrichment experiment in south-east Wyoming, USA. It is not clear how this study focusing on a semi-arid and strong continental climate with hot summers and little precipitation, is applicable to the climates of the northern high latitudes.

1.3 For example, given that high-latitude systems (1) are often strongly N limited; (2) CO₂ fertilization requires additional N supply; and (3) none of the tested models even represent nutrient constraints, it is hard to believe they are even predicting the current constraints on GPP reasonably, much less the future responses.

Given this, and other, evidence, I do not find it believable that the CMIP5 simulations can be used to infer high-latitude GPP sensitivity to climate over decades.

- Four out of seven models in our study represent nutrients constraint, where CESM1-BGC- and NorESM1-ME include an explicit nitrogen cycle (Page 7, Lines 6-16). The implementation resulted in an overly strong limitation of the CO₂ fertilization effect. The HadGEM2-ES model overestimates the increase in GPP most likely due to lacking integration of carbon-nitrogen dynamics. The spread among the model ensemble is essential for the Emergent Constraint approach. Extended Data Table 2 provides information about the specific configurations of each model, such as the implementation of the nitrogen cycle. The referee actually refers to a study using CESM which includes nitrogen dynamics (Bouskill et al., 2014).

- In addition, the magnitude of nitrogen limitation of the CO₂ fertilization effect is rather uncertain. We implemented the nitrogen limitation into the CMIP6 version of MPI-ESM and found that the effect is rather minor, mainly because biological nitrogen fixation could increase in response to the plant demand (Goll et al., 2017). Moreover, warming of permafrost soils might release nutrients from thawed organic material and thereby fertilize the system. This aspect was already discussed in our responses in the first round of reviews.
- We included a paragraph discussing the importance of carbon-nitrogen interactions in CMIP5 models to establish clarity on that matter (Page 7, Lines 6-16).

2.1 That extrapolation must be linear with respect to the independent variable being used (i.e., ?). Although the models show this linearity, there is no demonstration that the real world follows a linear relationship. Current understanding of high-latitude system dynamics indicates that these ecosystems will respond non-linearly to potentially dramatic changes in geomorphology, hydrology, plant distributions, fire, and nutrient availability.

- The idea of Emergent Constraints is to use a variable that is observable on a large scale to constrain another feature that cannot be observed accurately or not at all. The Emergent Constraint concept does not require, that the relationship between the changes in the observable predictor (e.g. LAI) and changes in the non-observable predictand (e.g. GPP) is linear for each model or in reality. The essential point is that the relationship is linear within the multi-model ensemble, meaning the models agree on the dynamics. For instance, Cox et al. (2018) recently elaborated a linear Emergent Constraint estimate on climate sensitivity at 2xCO₂ (predictand) analyzing historical temperature data (predictor) for the CMIP5 ensemble, although temperature does not relate linearly to CO₂. Here we demonstrate that there is a strong linearity between the predictor and predictand up to 2xCO₂ in each model, which becomes nonlinear with higher CO₂ levels (Extended Data Figure 3). But all models agree on the nonlinear aspect of saturation between changes in LAI and changes in GPP at concentrations higher than 2xCO₂. This agreement in all models results in the preservation of the, although weaker, linear relationship across the ensemble even for higher CO₂ concentration. The weakening of the linear relationship shows that the models gradually diverge at very high CO₂ concentration (4xCO₂) – a discussion of this aspect is beyond the scope of our current study. We acknowledge the complexity of the method and the interpretation of the behavior of a multi-model ensemble. We are currently drafting another manuscript that sheds light on this complex subject matter.
- The referee questions the linear relationship between changes in LAI and GPP in the real world. To address this concern, we analyzed all available FLUXCOM datasets of upscaled eddy covariance flux measurements for northern high latitudes GPP. These datasets were designed not to capture long-term changes as well as interannual variability and thus cannot be applied for a temporal analysis (e.g. Anav et al., 2015). But one can build on the spatial information to investigate the correlation between LAI and GPP. Using the climatologic mean of the recommended ensemble median of all FLUXCOM datasets and two independent sets of satellite observed LAI, we find a striking linear relationship for the northern high latitudes (Fig. R4.1a,b). This tight linear relation between the two variables over a wide range of values suggests that changes in GPP also result in changes in LAI.

- To further assess the relationship between changes in GPP and LAI, we used *in-situ* flux measurements, although these records are yet too short. We selected the longest FluxNet time series existing for the northern high latitudes, Hyytiälä, Finland (61.8474° N, 24.2948° E, 1996 - 2014). Further we took the surrounding pixels of the long-term but rather coarse (AVHRR, 1/12°, 1982 - 2016) as well as short-term but higher resolution (MODIS, 500m, 2000 - 2016) satellite observations of LAI. We find contemporary trends in GPP and LAI, but the linear relation between the *in-situ* measured GPP and long-term AVHRR satellite datasets is rather weak due to the coarse resolution. Thus, to match the flux tower footprint, we have to make recourse to high resolution satellite observations of MODIS. MODIS LAI and AVHRR LAI (the latter used in our study) have strong correlation and the latest AVHRR LAI datasets were developed by referencing to MODIS LAI (Zhu et al., 2013). For the MODIS time-series we find a much stronger relationship to the flux measurements and therefore confirms the tight connection between changes in GPP and LAI for the NHL (Fig. R4.1c). However, the overlapping period of MODIS and FluxNet is yet too short to derive reliable estimates. Anav et al. (2015) also analyzed other eddy covariance flux measurement sites and find a general agreement on increasing GPP.

Figure R4.1: Strong correlation in the climatologic mean in observational datasets between LAI_{max} derived from two independent satellite sensors, MODIS (a) and AVHRR (b), and the ensemble median annual average GPP from the FLUXCOM ensemble for the northern high latitudes. Color density indicates the probability distribution estimated using Gaussian kernel. c, Contemporary trends in the longest *in-situ* GPP flux measurement record in the NHL and the study site surrounding pixels of high resolution LAI satellite observations. The blue line shows the best linear fit and the shading shows the 95% confidence interval.

2.2 Given that the CMIP5 models clearly have unrealistic representations of high-latitude processes, using them to make an argument about linearity seems inappropriate.

- The CMIP5 models do well simulate physical and biogeochemical processes in high latitudes relevant to this study (our response to comment 1.3). The quality of simulations is evaluated in many other studies. As stated earlier, ESMs are not perfect. They, however, remain the most powerful tools available to predict future changes. The strength of the EC approach is that it does not require that models are correct in the quantification of relevant processes but that they encapsulate the correct trajectory.
- Linearity in each model is not required by the EC method (our response to comment 2.1).

3.1 given all the complexity of linking changes in NDVI to changes in LAI to changes in GPP, especially for high latitude systems;

- The LAI data used in our study have been validated (Zhu et al., 2013) and used widely (Poulter et al., 2014; Anderegg et al., 2015; Keenan et al., 2016; Mao et al., 2016; Zhu et al., 2016; Zeng et al., 2017; amongst others). While analyzing interannual variations in two of the AVHRR based LAI data sets, Forzieri et al. (2017) found that the data set used in our study has an improved harmonization across multiple sensor data streams and that the other LAI data set shows an artificial jump (about 50% increase) from improper harmonization of multi-sensor records. The LAI data set used in our study is the best available long-term LAI data set currently, and as mentioned above, is widely used in many previous studies.
- An independent study, Zhang et al. (2017), reported matching greening trends between the latest (Version 6) MODIS (a reference or benchmark data set) and AVHRR (Version 3) vegetation index data sets.
- Please see the analysis on observed relation between GPP and LAI presented in response to comment 2.1.

3.2 no assessment of the uncertainties in this calculation; and

- The LAI data used in our study has been extensively assessed in Zhu et al., 2013 (please see Section 3 of this article).
- Uncertainty in the relation between GPP and LAI is assessed using bootstrapping method. Please see Methods §8 in the manuscript.

3.3 no demonstration using any ground observations to test whether the hypothesized links between GPP and are accurate, it does not seem reasonable to conclude that the calculated GPP versus relationship is sufficiently robust to justify an extrapolation to the future.

- Please see our response to Comment 2.1 above.

References:

Anav, A., Friedlingstein, P., Beer, C., Ciais, P., Harper, A., Jones, C., Murray-Tortarolo, G., Papale, D., Parazoo, N.C., Peylin, P., Piao, S., Sitch, S., Viovy, N., Wiltshire, A., Zhao, M.: Spatiotemporal patterns of terrestrial gross primary production: A review. *Rev. Geophys.* 53, 2015RG000483, 2015.

Anderegg, W. R. L., Ballantyne, A. P., Smith, W. K., Majkut, J., Rabin, S., Beaulieu, C., ... Pacala, S. W. (2015). Tropical nighttime warming as a dominant driver of variability in the terrestrial carbon sink. *Proceedings of the National Academy of Sciences*, 112(51), 15591–15596.

Bouskill, N. J., Riley, W. J., and Tang, J. Y.: Meta-analysis of high-latitude nitrogen-addition and warming studies implies ecological mechanisms overlooked by land models, *Biogeosciences*, 11, 6969-6983, doi:10.5194/bg-11-6969-2014, 2014.

Cox, P. M., Huntingford, C., & Williamson, M. S. (2018). Emergent constraint on equilibrium climate sensitivity from global temperature variability. *Nature* 553, 319

De Kauwe, M. G., Medlyn, B. E., Walker, A. P., Zaehle, S., Asao, S., Guenet, B., Harper, A. B., Hickler, T., Jain, A. K., Luo, Y. Q., Lu, X. J., Luus, K., Parton, W. J., Shu, S. J., Wang, Y. P., Werner, C., Xia, J. Y., Pendall, E., Morgan, J. A., Ryan, E. M., Carrillo, Y., Dijkstra, F. A., Zelikova, T. J., and Norby, R. J.: Challenging terrestrial biosphere models with data from the long-term multifactor Prairie Heating and CO₂ Enrichment experiment, *Global Change Biology*, 23, 3623-3645, 2017.

Forzieri, G., Alkama, R., Miralles, D. G., & Cescatti, A. (2017). Satellites reveal contrasting responses of regional climate to the widespread greening of Earth. *Science*, 356(6343), 1180–1184.

Goll, D.S., Winkler, A.J., Raddatz, T., Dong, N., Prentice, I.C., Ciais, P., Brovkin, V.: Carbon–nitrogen interactions in idealized simulations with JSBACH (version 3.10). *Geosci. Model Dev.* 10, 2009–2030, 2017.

He, Y. J., Trumbore, S. E., Torn, M. S., Harden, J. W., Vaughn, L. J. S., Allison, S. D., and Randerson, J. T.: Radiocarbon constraints imply reduced carbon uptake by soils during the 21st century, *Science*, 353, 1419-1424, 2016.

Keenan, T. F., Prentice, I. C., Canadell, J. G., Williams, C. A., Wang, H., Raupach, M., & Collatz, G. J. (2016). Recent pause in the growth rate of atmospheric CO₂ due to enhanced terrestrial carbon uptake. *Nature Communications*, 7, 13428.

Koven, C. D., Riley, W. J., and Stern, A.: Analysis of Permafrost Thermal Dynamics and Response to Climate Change in the CMIP5 Earth System Models, *J Climate*, 26, 1877-1900, Doi 10.1175/Jcli-D-12-00228.1, 2013.

Mao, J., Ribes, A., Yan, B., Shi, X., Thornton, P. E., Séférian, R., ... Lian, X. (2016). Human-induced greening of the northern extratropical land surface. *Nature Climate Change*, 6(10), 959–963. <https://doi.org/10.1038/nclimate3056>

Poulter, B., Frank, D., Ciais, P., Myneni, R. B., Andela, N., Bi, J., ... van der Werf, G. R. (2014). Contribution of semi-arid ecosystems to interannual variability of the global carbon cycle. *Nature*, 509(7502), 600–603.

Todd-Brown, K. E. O., Randerson, J. T., Post, W. M., Hoffman, F. M., Tarnocai, C., Schuur, E. A. G., and Allison, S. D.: Causes of variation in soil carbon simulations from CMIP5 Earth system models and comparison with observations, *Biogeosciences*, 10, 1717-1736, Doi 10.5194/Bg-10-1717-2013, 2013.

Zaehle, S., Medlyn, B. E., De Kauwe, M. G., Walker, A. P., Dietze, M. C., Hickler, T., Luo, Y. Q., Wang, Y. P., El-Masri, B., Thornton, P., Jain, A., Wang, S. S., Warlind, D., Weng, E. S., Parton, W., Iversen, C. M., Gallet-Budynek, A., McCarthy, H., Finzi, A. C., Hanson, P. J., Prentice, I. C., Oren, R., and Norby, R. J.: Evaluation of 11 terrestrial carbon-nitrogen cycle models against observations from two temperate Free-Air CO₂ Enrichment studies, *New Phytologist*, 202, 803-822, 2014.

Zeng, Z., Piao, S., Li, L. Z. X., Zhou, L., Ciais, P., Wang, T., ... Wang, Y. (2017). Climate mitigation from vegetation biophysical feedbacks during the past three decades. *Nature Climate Change*, 7(6), 432.

Zhang, Y., Song, C., Band, L. E., Sun, G., & Li, J. (2017). Reanalysis of global terrestrial vegetation trends from MODIS products: Browning or greening? *Remote Sensing of Environment*, 191, 145–155.

Zhu, Z., Bi, J., Pan, Y., Ganguly, S., Anav, A., Xu, L., ... Myneni, R. B. (2013). Global Data Sets of Vegetation Leaf Area Index (LAI)_{3g} and Fraction of Photosynthetically Active Radiation (FPAR)_{3g} Derived from Global Inventory Modeling and Mapping Studies (GIMMS) Normalized Difference Vegetation Index (NDVI)_{3g} for the Period 1981 to 2011. *Remote Sensing*, 5(2), 927–948.

Zhu, Z., Piao, S., Myneni, R. B., Huang, M., Zeng, Z., Canadell, J. G., ... Zeng, N. (2016). Greening of the Earth and its drivers. *Nature Climate Change*, 6(8), 791–795.

Authors Response to Referee 2 (Nature Communications manuscript 17-32151-T)

This manuscript uses the emergent constraints concept to estimate the change in northern-hemisphere land GPP in response to a doubling of CO₂ in comparison to pre-industrial conditions. The study has a strong similarity to a previous study by Wenzel et al. (2016). The main difference is that this study uses satellite observations of leaf area index to derive the emerging constraint on the GPP change. This is a novel approach and in my view a major advance in comparison to Wenzel et al. (2016). The study further provides three other evidences for GPP change that are independent of the LAI data in order to support their main estimate. The study concludes that Earth system models underestimate the change in carbon uptake in response to CO₂ change.

- We thank Dr. Forkel for his open and detailed review. The revisions done in response to these comments has resulted in an improved manuscript.

I was asked to comment on the revisions and whether the reviewers' concerns were adequately addressed. In my view, the responses to the reviewers were adequate. However, some of the responses are only partly reflected in the revised manuscript. I'm convinced that the manuscript is suited for publication in Nature Communications but I think the following changes are necessary:

1 Derivation and scaling of NPP estimates

The main conclusion (= title) of the study is “Earth System Models Underestimate Carbon Fixation by Plants in the High Latitudes”. However, only seven models are used, one model (HadGEM2-ES) overestimates Delta-GPP, one models are likely within the uncertainty of the observational-derived Delta-GPP (MIROC-ESM, MPI-ESM-LR), and only four models underestimate Delta-GPP (Fig. 2c). Also the probability density functions of the Delta-GPP distribution from the emerging constrain and from the multi-model mean largely overlap. So the question is, if this difference is really significant (n = 7 models!) to conclude that “Earth System Models Underestimate Carbon Fixation by Plants in the High Latitudes”. For sure, you can conclude that some models underestimate. However, it does not matter if ESMS – in average – under- or overestimate land carbon uptake. The results nicely describe the variety of LAI, LAI changes, LAI-PCI sensitivities in the models but then the remaining text focusses on conclusions about the multi-model mean, which has likely a very limited meaning if the models show such a variety. What really matters is how individual models perform and which processes contribute to an over- or underestimation. In order to provide a new insight, the text (and title) should either focus on 1) the novel approach to use LAI data as an emergent constraint for GPP changes or 2) an analysis and discussion about land-atmosphere processes and feedbacks that affect GPP change in the observational world and in the world of each model. Such a discussion should cover potential impacts on GPP through changes in seasonality/phenology, nutrient availability, disturbances and vegetation dynamics (increase in tree and shrub cover), permafrost and water availability (reviewer 2).

- Below, each point raised by the referee is addressed.

*1.1 Change the title and adapt the scope of the study from “Models Underestimate ...” to the presentation of a new EC-approach to estimate Delta-GPP/2*CO₂ (see Comment 1).*

- Earth system models that include an interactive global carbon cycle are capable of simulating the interface and potential feedbacks between terrestrial, oceanic, and atmospheric processes. This setup is needed to investigate on how the entire system might

change under ongoing forcings such as CO₂ emissions. All of these models which provided output for the required CMIP5 simulations are analyzed in this study. This set of CMIP5 models is also used in several other studies to investigate on carbon-climate feedbacks and climate projections for different scenarios of CO₂ emission pathways (Arora et al., 2013, Anav et al., 2013, Friedlingstein et al., 2013, Wenzel et al., 2014, Wenzel et al., 2016, amongst others). Conventionally, the multi-model mean is the metric most conclusions are based on. These again serve as a basis for the IPCC report, and thus, for policy makers.

- This study shows that more sophisticated concepts are needed and contributes to the ongoing discussion on challenges of combining climate projections in a multi-model ensemble (Knutti et. al, 2009, 2010, 2017).
- Moreover, the focus of the manuscript essentially contributes to the ongoing discussion on whether ESMs are over- or underestimating photosynthetic CO₂ fixation based on the small sample-size CMIP5 ensemble (Friedlingsstein et al., 2013, Smith et al., 2016, De Kauwe et al., 2016, Campbell et al., 2017, amongst others), e.g.:
 - Friedlingstein et al. (2013) found that most of the carbon emission driven models are overestimating present-day atmospheric CO₂ concentration and identified the land carbon sink to be the highest source of uncertainty for climate projections. Our study shows that the average overestimation of atmospheric CO₂ concentration potentially arises from underestimation of the increase in photosynthetic carbon fixation of most models and helps to reduce the uncertainty in the land carbon sink.
 - Smith et al. (2016) also analyzes CMIP5 models and satellite data to find that ESMs generally are overestimating GPP. We show that the opposite is the case for most models using different sets of observations.
- In summary, we find that the manuscript's focus on the underestimation of photosynthetic carbon fixation of most models is the most important aspect. We agree with the referee that to advance scientific model development light has to be shed on the origins of underestimation. Consequently, this section has been rewritten and the causes are assessed and described in more detail (Page 4, Lines 6-17; Page 7, Lines 6-16).

1.2 Include also a short discussion of reasons (processes) that contribute to the fact that some models over-/underestimate the changes (Comment 1).

- We expanded the discussion on the origins of over- and underestimation in (Page 4, Lines 6-17; Page 7, Lines 6-16). Please see response to comment 1.1.

2 Derivation and scaling of NPP estimates

The author present an estimate of NPP change in the NHL latitudes as an independent estimate or evidence for their findings. To estimate NPP, the annual minimum of the seasonal cycle of CO₂ is converted to a change in land carbon flux (chapter 5 in Methods). Reviewers 1 and expressed their concern about this approach.

I'm not convinced by this approach. First, this approach is presented without any references. The authors provide no evidence that the approach is valid. The used terminology is also not well defined. Delta-F_C,BRW is estimated from CO₂ minimum. Delta-F_NPP,EC is

estimated from Delta-F_GPP,EC. Which metrics are then used and compared in the main text (P5 L10-22)?

Second, the approach is based on several assumptions which are definitely not true: 1) the change in the CO₂ minimum is only affected by NPP, 2) a constant conversion factor of CO₂ fractions (ppm) to carbon flux (PgC), and 3) a constant contribution of high latitude areas to the CO₂ minimum. Various studies investigated the influencing factors on the seasonality of CO₂ (Randerson et al., 1997; Graven et al., 2013; Forkel et al., 2016). From these studies, we know that

1) changes in CO₂ seasonality are affected by productivity and respiration (less by other carbon fluxes),

2) that the contribution of GPP and ecosystem respiration to the CO₂ amplitude changes over time (e.g. Tab S6 in (Forkel et al., 2016)), and

3) the contribution of different regions (e.g. Arctic or boreal region) also changed over time depending on the magnitude of the regional changes in land carbon uptake. For example, arctic regions contribute around 26% and 32% and boreal regions 54 and 52% to the seasonal amplitude of CO₂ at BRW in the 1970s and 2000s, respectively (Tab. S5 in (Forkel et al., 2016)).

In summary, this approach provides more questions than it provides a robust and reliable independent estimate of the emerging constraint. The approach does not at all support the findings of the study but has for me the potential to destroy the entire flow of the paper. Please remove these estimates from the paper.

2.1 Remove the estimation of NPP changes (evidence approach 1). This approach raises more questions and does not provide a reliable evidence for the LAI-based estimates (Comment 2). However, it is a good idea and likely possible to use the seasonality of CO₂ as constraint for GPP. For example, a relationship between changes in GPP and changes in CO₂ amplitude is presented in Figure 3 in Forkel et al. (2016).

- We accept these comments and removed this line of evidence. Alternatively, we followed the suggestion by the reviewer to use the seasonality of atmospheric CO₂ at high latitudinal stations to seek for further confirmation of the EC estimate (Page 6, Lines 1-19; Methods §5; Fig. 3c). Please see response to comment 4.23 for more detail.

*2.2 Furthermore GPP products that were upscaled from eddy covariance observations (Tramontana et al., 2016) can be used to derive the Delta-GPP/LAI_max sensitivities from more reliable data-driven estimates. Such a dataset also has limitations. For example, it is unclear how well it reproduces the CO₂ fertilization effect and of course it also does not “see” a period with 2*CO₂ increase. However, I’m still convinced that such an approach is more reliable than estimating Delta-NPP from the CO₂ minimum.*

- We followed the advice by the reviewer and analyzed all available FLUXCOM datasets of upscaled eddy covariance flux measurements for northern high latitudes GPP. For details, please see response to comment 3.2 and Figure R5.3.

- However, it is also known that these upscaled FLUXCOM products are not to be used for trend or interannual variability analysis (Anav et al. 2015, Jung et al. 2009, Anav et al. 2013, Piao et al. 2013), and thus, we decided not to include these results in the manuscript.

3. Include MODIS-based EC estimates and compare multiple EC estimates

3.1 Reviewer 1 mentioned that the results could be affected by the uncertainty in satellite estimates of LAI. When I was reading the manuscript (before reading the reviewer comments and responses), I had the same question. The authors provided a good response to the reviewer comment. However, this is not appropriately reflected in the revised manuscript. Figure R1.2 should be included in the manuscript (Extended Data). Moreover, the MODIS-based estimates should be also included Fig. 2 to demonstrate the effect of data uncertainty in the derived EC estimates.

- Yes, the analysis was also conducted using MODIS data. The MODIS greening sensitivity is within the range of uncertainty of the AVHRR long-term greening sensitivity and on average lower by 13% (Fig. R5.1). This linearly translates to a 13% lower estimate of ΔGPP at $2\times\text{CO}_2$. Two caveats are in order. First, the MODIS time period (16 years) is too short to provide a reliable estimate of greening sensitivity and is thus not comparable to more robust estimates from longer time periods from CMIP5 and AVHRR (see high variability of 15-year greening sensitivities in Fig. R5.2). Second, the overlap between historical forcing used in CMIP5 and the MODIS record is quasi not given. The comparability between models and observations is an essential aspect in the EC approach and must be ascertained to provide reliable estimates. For instance, using short-term observed sensitivity at a higher forcing level as a constraint on the multi-model linear relationship based on long-term modelled sensitivities at a lower forcing level results in an incorrect EC estimate. We are drafting a manuscript that deals with these issues, methodological caveats, and potential erroneous conclusions when using the EC method.

Figure R5.1: LAI_{\max} sensitivity to ω . LAI_{\max} data are from AVHRR and Terra MODIS sensors. ω is the first principal component of annual mean atmospheric CO_2 and annual growing degree days above 0°C .

Figure R5.2: Temporal variation of LAI_{\max} sensitivity to ω of three selected CMIP5 models spanning the full range from low (CESM1-BGC, **a**), to closest-to-observations (MIROC-ESM, **b**), to high-end (HadGEM2-ES, **c**) LAI_{\max} sensitivity. The colored lines show LAI_{\max} sensitivity variations for moving windows of varying length of 15 (blue), 30 (green), and 45 (red) years over the historical period from 1860 to 2005.

3.2 As the study provides four different estimates of Delta-GPP and it is an improved follow-up of the work by Wenzel et al. (2016), I advise you to show all recent estimates in one plot. The Delta-GPP/2*CO2 estimates from LAI3g and MODIS, from the ESMs, from Wenzel et al. (2016), from the upscaling GPP, and the Delta-NBP/2*CO2 estimates from the inversions should be all shown in one plot (barplot with error estimates). This figure will provide insight in the current uncertainty and the most likely values of Delta-GPP/2*CO2 and will also help to compare your results with the results from Wenzel et al. (2016). Replace the current Figure 3 with this new figure.

Include a comparison of recent Delta-GPP/2*CO2 estimates (e.g. using MODIS) and replace Figure 3 with a new Figure (Comment 3).

- The comparison to Wenzel et al. (2016) is not possible because that study looked into idealized cases of CO₂ fertilization only and relative changes, i.e., the relative increase in GPP with respect to the preindustrial state.
- As suggested, we compiled a comparison bar plot for all Δ GPP (FLUXCOM, ESMs) respectively Δ NBP (CO₂ inversion) estimates scaled to an CO₂ increment of 1 ppm (Fig. R5.3).
- We decided not to include it in the main manuscript for two reasons. First, the FLUXCOM datasets are not designed to be analyzed with respect to long-term changes and, thus, the meaning of the comparison to other estimates is limited. Second, CO₂ inversions provide estimates for NBP which can only be put into comparison to the EC Δ GPP estimate making some assumptions. This is already covered in the manuscript (Page 6, Line 21 – Page 7, Line 4). Consequently, this comparison plot does not add more information than that already in the text.

Figure R5.3: Comparison of different Δ GPP (FLUXCOM, ESMs) respectively Δ NBP (CO₂ inversion) estimates scaled to an CO₂ increment of 1 ppm.

4 Clearly define and consistently use your terms. It is often not clear if you mean the sensitivity of LAI_max to CO2+GDD0 or the sensitivity of Delta-GPP to LAI, or if you mean GPP, NPP, or the overall land carbon uptake (NBP) (see my Specific Comments).

- Please see below each specific comment addressed with an individual response.

4 Specific Comments:

4.1 (P2 L1-3): Please provide a reference for any numbers that are not yours.

- Numbers have been updated according to the recently released Global Carbon Budget report 2017 and respective reference has been added (Page 2, Lines 1-3).

4.2 (P2 L3-6): *I propose to split this sentence into two (e.g. after “regions”) to improve readability.*

- The passage has been modified as suggested (Page 2, Lines 5-6).

4.3 (P2 L9): *Reference 13 (Forkel et al., 2014) does not fit to this sentence; throughout the paper this reference does not fit. Probably you intended to refer to my other paper: (Forkel et al., 2016). I also propose to place reference 13 together with references 14 and 15.*

- The references have been updated in the manuscript as suggested by the referee (Page 2, Line 10; Page 3, Line 2; Page 4, Line 1; Page 5, Line 19; Page 6, Line 2).

4.3 (P3 L30): *Can you refer to a figure where the linear relation can be seen? – Figure 2a? The sentence does not really fit as a topic sentence for this paragraph, but it rather describes a result and hence should come later. It would even better fit as a summary sentence of this paragraph. You should rather start the paragraph with an introductory sentence about the general sensitivity of LAI_max to CO2 and temperature (or GDD).*

- This paragraph has been revised following the suggestions of the referee (Page 3, Line 30 – Page 4, Line 1).

4.4 (P4 L5): *“span a large range in a somewhat complex manner” – This expression can mean anything. I don’t understand what you want to say here and how this links with Fig. 2 a and b. Please be more specific.*

- This passage has been rewritten to be more precise (Page 4, Lines 4-6).

4.5 (P4 L1-16): *In order to make the text better understandable, I propose to define the term greening “rate” or “sensitivity” explicitly already at the beginning and then use one of the two terms consistently throughout the manuscript. However, I understand “rate” rather as a change in time. So I would use “greening rate” for temporal trends in LAI_max, “greening sensitivity” for the slope between LAI_max and PCI, and “LAI_max-GPP sensitivity” for slopes between Delta-LAI_max and Delta-GPP.*

- The manuscript has been updated following the suggestion by the referee. Now the terminology is precisely defined and used consistently throughout the text (Page 3, Line 8; Page 4, Line 5-6, 7, 13, 16-17, 22, 24, 27).

4.6 (P4 L21-23): *It is not clear in this sentence if you mean the sensitivity of LAI_max to PCI (as suggested by the reference to Fig 2) or to Delta-GPP (as suggested by the references to Extended Data Figures 2 and 3 in the previous sentence). This can be solved if you define the terms already at the beginning (see previous comment).*

- Please see the response to previous comment.

4.7 (P4 L25-27): *“linear trend”: Do you mean linear “relation”? I associate “trends” rather with temporal changes.*

- ‘Trend’ has been replaced by ‘relation’ (Page 4, Line 26).

4.8 (P5 L13): *You can also add the updated reference 13.*

- The reference has been updated. Please see response to ‘Specific Comment’ 4.3 for details.

4.9 (P5 L14-16): You make a very big step in this sentence! Although you refer to Methods §5, it is crazy to just jump from CO2 minimum changes to a change in NPP. For me as a reader, this just creates a feeling like “Are you joking? Now I don’t trust you”. You really should add here a sentence how you get to this NPP change.

- The sections concerning this line of evidence has been removed from the revised manuscript as suggested by the reviewer.

4.10 (P5 L17-18): NPP is a flux, i.e. it is per time (e.g. x PgC yr-1). If one describes a change in NPP per time (e.g. per year) the unit is PgC yr-2. I’m not certain how the “NPP enhancement” was calculated and what the 0.46 PgC yr-1 really mean. Should the unit be PgC yr-2? If not is this then an absolute change between 1960s and 2010s?

- Yes, the number represents the absolute change, so it shows that the carbon flux due to NPP increased by 0.46 Pg C yr⁻¹ comparing the 1960s with the 2010s. Thus, the correct unit is Pg C yr⁻¹. Anyway, this line of evidence will not be part of the revised manuscript anymore.

4.11 (P5 L19-20): I don’t understand the sentence. NPP is independent of Rh. But NBP or NEE include Rh (Chapin et al., 2006; Schulze, 2006):

$$NPP = GPP - Ra$$

$$NBP = (Rh + Ra + FireEmissions + other) - GPP$$

$$NEE = (Rh + Ra) - GPP$$

If Rh increased, why should the NPP trend be larger? Changes in respiration contributed to a decrease of the CO2 amplitude at BRW because respiration in northern latitudes happens more during summer because of higher temperatures. However, GPP increases contributed to a much larger increase in CO2 amplitude than respiration caused a decrease, which overall results in an increasing CO2 amplitude (Tab S6 in (Forkel et al., 2016)). These results likely also apply to the CO2 minima because changes in the CO2 amplitude are largely caused by changes in CO2 minima: An increase in Rh contributes hence to an increase in CO2 minima.

- This analysis is solely based on the annual CO₂ minimum which is controlled by terrestrial net primary production (NPP) as shown by e.g. Forkel et al., 2016 or Graven et al., 2013. So, the decrease in the CO₂ minimum reflects an increase in NPP. We derive an estimate for how much NPP must have increased to explain this decrease in CO₂ minimum. We find that it points to the same direction as the EC estimate and buttresses the statement that NPP increase might be underestimated in most models. Now, increase in heterotrophic respiration (Rh) during the growing season (e.g. due to warming) would result in an increase of the annual CO₂ minimum. If this is the case, then NPP must have increased even more to counterbalance the increase in Rh. Thus, the derived value of NPP enhancement is a conservative estimate in comparison with the NBP estimates by the inversion products (also because fire emissions etc. are not considered).
- We do not claim that this method provides an estimate for actual NPP increase in the northern high latitudes. We rather show that a simple back-of-the-envelope calculation

based on long-term measurements of CO₂ concentration yields an estimate that is in the same ballpark as the greening-based Emergent Constraint value. However, we agree with the referee that the uncertainty of our simple calculation is too high and therefore decided to remove this line of reasoning as stated earlier.

- Please also see response to comment 2.2.

4.12 (P5 L28): Again the updated reference 13 would better fit.

- The reference has been updated. Please see response to ‘Specific Comment’ 4.3 for details.

4.13 (P6 L5-6): If you refer to results for the Alert site, show them! It does not make sense to make the reader curious about results and then “close the door” with “results not shown”. One can see this impolite behaviour in many studies but it is completely useless. That means: If you refer to results, show them (i.e. include Fig R1.4). If you don’t want to show them for brevity, don’t mention them.

- The results for Alert Nunavut site are now included into the manuscript (Fig. 3b, c; Page 5, Line 11-31; Page 6, 1-19).

4.14 (P6 L18-20): Again, this sentence does not fit. The inversions quantify the net land exchange (NBP) and do not provide information on the component fluxes NPP and Rh.

- See “Specific Comment” 4.11. Again, this analysis shows that the estimates derived from the inversion products are conservative approximations, thus, buttressing that ESMs are on average underestimating.
- For clearance, we rewrote this passage (Page 6, Line 21 – Page 7, Line 4).

4.15 (P6 L24-30): I’m missing a short discussion on what processes the models miss or which process are miss-represented in order to reproduce 1) the correct LAI changes, and 2) the GPP change.

- The discussion with respect to reasons for misrepresenting LAI and GPP changes in ESMs has been expanded (Page 4, Lines 6-17; Page 7, Lines 6-16).

4.16 (P11 Chapter 1.1): This chapter can be shortened because it mostly describes previous published studies about LAI and LAI product generation. I propose to write only the essential information for your study and what you did with the data.

- Referees of the Nature review (round 1) requested more details on applied observational LAI products. Accordingly, this part cannot be shortened.

4.17 (P12 L21-26): When I was reading the manuscript, I was wondering already if the results will be the same with MODIS data. Please don’t tell me that you don’t show it for brevity. I really want to see the EC estimate that is based on MODIS data. Include Figure R1.2 in the text and add the MODIS-based results to all figures of the main text.

- Please see response to comment 3.1.

4.18 (P15 L26 – P16 L10). *This is basically a linear regression. The text reads like Cox et al. would have invented linear regression. I think this chapter can be significantly shortened.*

- This passage has been rewritten (Page 16, Line 27).

4.19 (P18 L21 – P19 L19): *Why you need to calculate PDFs? You could also just show the uncertainty in the EC estimate based on the distribution of the computed EC values.*

- PDFs (Fig. 2d) clearly illustrate two essential aspects here, the reduction in uncertainty using the EC approach with respect to the conventionally used multi-model mean, and, the probability shift to a substantial higher value. We believe that presenting the results in this manner conveys these messages well.

4.20 (Figure 1a – referee probably meant Figure 2a -): *Please describe in the caption how LAI_max and PCI were spatially integrated (averaged?).*

- Caption has been updated accordingly (Page 24, Line 5).

4.21 (Figure 2c): *Most of the paper describes that the multi-model mean underestimates the emerging constraint. Therefore, I propose to show the multi-model mean in this figure to visualize the underestimation. Additionally, the uncertainty estimate (or standard error) of the observational-derived Delta-GPP estimate should be added as shaded area round the dashed line.*

- Figure 2d already shows the comparison between the multi-model mean and the observation-derived Delta-GPP estimate including the uncertainty estimate. The suggested changes to Figure 2c make it too crowded and limits the readability of the figure.

4.22 (Figure 2d): *I don't understand why the Delta-GPP estimate is shown as PDF. I would rather prefer to see estimates for individual models and multi-model mean. Please show the results as barplot like in Fig. 2 b because this shows just the data as it is and does not rely on the assumption of normal distribution. It would also help to read the manuscript because as a reader one is already used to the barplot. There is no need to mimic all the PDF-like figures from the Cox et al. and Wenzel et al. studies!*

- Please see our response to comment 4.19 for arguments in favor of presenting the results using PDFs.

4.23 (Figure 3): *Actually, I don't understand how these datasets provide evidence for the "emerging constraint estimate of NHL GPP" and how they are connected with the previous figures. These datasets all show an increase of the land carbon uptake which is however known from previous studies working with these datasets (Graven et al., 2013; Forkel et al., 2016; Wenzel et al., 2016). But where is the link to the emerging constraint on Delta-GPP? Reviewer 3 also mentions that this figure repeats just previous findings. Please move this figure to the supplement and replace it with a figure (e.g. barplot) that provides an overview of recent Delta-GPP/2*CO2 estimates (i.e. emerging constraints, inversion-based results, results based on CO2 amplitude).*

- Figure 3 has been substantially restructured. As recommended by the reviewer earlier, it now shows changes in the CO₂ amplitude at Point Barrow (Fig. 3a) as well as Alert Nunavut (Fig. 3b) station in comparison to CMIP5 models. Fig. 3c now shows a new approach using the slope of summertime CO₂ drawdown at these stations as a metric to

estimate high NHL GPP (Methods §5). We kept Fig. 3d - to our knowledge, this is the first time that a NHL vs. Arctic Ocean comparison of changes of the seasonality of CO₂ flux is presented using a high-resolution model ensemble. This figure clearly shows that the terrestrial processes determine the nature of the CO₂ seasonal cycle (Page 26 & 27).

- Figure showing the changes in the seasonal cycle of land-atmospheric CO₂ exchange estimated by two inversion procedures was moved to the supplement as suggested by the reviewer (Extended Data Fig. 5, Page 32).

4.24 (Extended Data Figure 2): Please indicate if this also integrates over the NHL regions.

- Caption of Extended Data Figure 2 has been changed accordingly (Page 29, Line 3).

*4.25 (Extended Data Figure 3): From the caption, I understand that blue is [2*CO₂] – [1*CO₂] but from the legend blue is “< 2*CO₂” which I don’t understand. So I don’t really understand what model outputs were used or selected to compute Delta_LAI and Delta_GPP for this figure. Please make the caption and legend consistent. Additionally, the dots and lines from Extended Data Figure 2 could be added to this plot which would reduce the number of figures.*

- Legend and caption have been modified to be consistent (Page 30, Line 3). ED Figure 2 and ED Figure 3 should not be combined. On the one hand, two conceptually different sets of simulations are illustrated (ED Figure 2: CMIP5 esmHistorical; ED Figure 3 CMIP5 1pctCO₂), and on the other hand, a combined figure would be quite crowded failing to clearly communicate the individual message of each figure.

4.26 (Extended Data Figure 4): The figure is very difficult to understand because the colours in the legend do not fit with shown symbols and lines. For example, red squares show “CO₂ fertilization” effect but what are then the squares in the other colours? Additionally, what purpose has the legend for the models? Bold lines with colours for each model are not in this plot. And the green fitting line is a fit through all crosses, even if they are not green-coloured? You need to definitely improve this plot (fit legend symbol/colours with the data), otherwise one is only puzzled with linking the symbols and this distracts completely from the message of the figure.

- The colors and legend in Extended Data Figure 4 have been improved following the suggestions above. Now, the figure conveys the message in a much more precise manner (Page 31).

4.27 (Extended Data Figure 5): You always say that the multi-model mean underestimates the EC estimate. So can you add a cross for the multi-model mean to show this underestimation?

- The multi-model mean estimate is now indicated in the (now) Extended Data Figure 6 (Page 33).

References:

- Anav, A., Murray-Tortarolo, G., Friedlingstein, P., Sitch, S., Piao, S., & Zhu, Z. (2013). Evaluation of Land Surface Models in Reproducing Satellite Derived Leaf Area Index over the High-Latitude Northern Hemisphere. Part II: Earth System Models. *Remote Sensing*, 5(8), 3637–3661.
- Anav, A., Friedlingstein, P., Beer, C., Ciais, P., Harper, A., Jones, C., ... Zhao, M. (2015). Spatiotemporal patterns of terrestrial gross primary production: A review. *Reviews of Geophysics*, 53(3), 2015RG000483.
- Arora, V. K., Boer, G. J., Friedlingstein, P., Eby, M., Jones, C. D., Christian, J. R., ... Wu, T. (2013). Carbon–Concentration and Carbon–Climate Feedbacks in CMIP5 Earth System Models. *Journal of Climate*, 26(15), 5289–5314.
- Campbell, J. E., Berry, J. A., Seibt, U., Smith, S. J., Montzka, S. A., Launois, T., ... Laine, M. (2017). Large historical growth in global terrestrial gross primary production. *Nature*, 544(7648), 84–87.
- Chapin, F.S., Woodwell, G.M., Randerson, J.T., Rastetter, E.B., Lovett, G.M., Baldocchi, D.D., Clark, D.A., Harmon, M.E., Schimel, D.S., Valentini, R., Wirth, C., Aber, J.D., Cole, J.J., Goulden, M.L., Harden, J.W., Heimann, M., Howarth, R.W., Matson, P.A., McGuire, A.D., Melillo, J.M., Mooney, H.A., Neff, J.C., Houghton, R.A., Pace, M.L., Ryan, M.G., Running, S.W., Sala, O.E., Schlesinger, W.H. & Schulze, E.-D. (2006) Reconciling Carbon-cycle Concepts, Terminology, and Methods. *Ecosystems*, 9, 1041–1050.
- De Kauwe, M. G., Keenan, T. F., Medlyn, B. E., Prentice, I. C., & Terrer, C. (2016). Satellite based estimates underestimate the effect of CO₂ fertilization on net primary productivity. *Nature Climate Change*, 6(10), 892–893.
- Forkel, M., Carvalhais, N., Rödenbeck, C., Keeling, R., Heimann, M., Thonicke, K., ... Reichstein, M. (2016). Enhanced seasonal CO₂ exchange caused by amplified plant productivity in northern ecosystems. *Science*, 351(6274), 696–699.
- Forkel, M., Carvalhais, N., Schaphoff, S., v. Bloh, W., Migliavacca, M., Thurner, M. & Thonicke, K. (2014) Identifying environmental controls on vegetation greenness phenology through model–data integration. *Biogeosciences*, 11, 7025–7050.
- Friedlingstein, P., Meinshausen, M., Arora, V. K., Jones, C. D., Anav, A., Liddicoat, S. K., & Knutti, R. (2013). Uncertainties in CMIP5 Climate Projections due to Carbon Cycle Feedbacks. *Journal of Climate*, 27(2), 511–526.
- Graven, H.D., Keeling, R.F., Piper, S.C., Patra, P.K., Stephens, B.B., Wofsy, S.C., Welp, L.R., Sweeney, C., Tans, P.P., Kelley, J.J., Daube, B.C., Kort, E.A., Santoni, G.W. & Bent, J.D. (2013) Enhanced Seasonal Exchange of CO₂ by Northern Ecosystems Since 1960. *Science*, 341, 1085–1089.
- Knutti, R., Furrer, R., Tebaldi, C., Cermak, J., & Meehl, G. A. (2009). Challenges in Combining Projections from Multiple Climate Models. *Journal of Climate*, 23(10), 2739–2758.
- Knutti, R. (2010). The end of model democracy? *Climatic Change*, 102(3–4), 395–404.

Knutti, R., Sedláček, J., Sanderson, B. M., Lorenz, R., Fischer, E. M., & Eyring, V. (2017). A climate model projection weighting scheme accounting for performance and interdependence. *Geophysical Research Letters*, 44(4), 1909–1918.

Kolby Smith, W., Reed, S. C., Cleveland, C. C., Ballantyne, A. P., Anderegg, W. R. L., Wieder, W. R., ... Running, S. W. (2016). Large divergence of satellite and Earth system model estimates of global terrestrial CO₂ fertilization. *Nature Climate Change*, 6(3), 306–310.

Piao, S., Sitch, S., Ciais, P., Friedlingstein, P., Peylin, P., Wang, X., ... Zeng, N. (2013). Evaluation of terrestrial carbon cycle models for their response to climate variability and to CO₂ trends. *Global Change Biology*, 19(7), 2117–2132.

Randerson, J.T., Thompson, M.V., Conway, T.J., Fung, I.Y. & Field, C.B. (1997) The contribution of terrestrial sources and sinks to trends in the seasonal cycle of atmospheric carbon dioxide. *Global Biogeochemical Cycles*, 11, 535–560.

Schulze, E.-D. (2006) Biological control of the terrestrial carbon sink. *Biogeosciences*, 3, 147–166.

Tramontana, G., Jung, M., Schwalm, C.R., Ichii, K., Camps-Valls, G., Ráduly, B., Reichstein, M., Arain, M.A., Cescatti, A., Kiely, G., Merbold, L., Serrano-Ortiz, P., Sickert, S., Wolf, S. & Papale, D. (2016) Predicting carbon dioxide and energy fluxes across global FLUXNET sites with regression algorithms. *Biogeosciences*, 13, 4291–4313.

Wenzel, S., Cox, P. M., Eyring, V., & Friedlingstein, P. (2014). Emergent constraints on climate-carbon cycle feedbacks in the CMIP5 Earth system models. *Journal of Geophysical Research: Biogeosciences*, 119(5), 794–807.

Wenzel, S., Cox, P.M., Eyring, V. & Friedlingstein, P. (2016) Projected land photosynthesis constrained by changes in the seasonal cycle of atmospheric CO₂. *Nature*, 538, 499–501.

Authors Response to Referee 3 (Nature Communications manuscript 17-32151-T)

This is a carefully executed analysis which significantly adds to the literature using emergent constraints to get more reliable estimates of key quantities in the global carbon cycle. I am satisfied with the authors' in-depth responses to the previous review comments.

- We thank Prof. Prentice for an open review of our manuscript and sharing the opinion that it adds significantly to the field of constraining key aspects of the global carbon cycle.

For my part I have only a few points to add:

*1. I still do not understand why there is a discrepancy between results of the EC approach and the conclusion of Thomas et al. that *all* models (in the MsTMIP ensemble) underestimate the historical response of LUE to CO₂. The present MS seems to indicate that it is just a matter of a different set of models. But Graven et al.'s (2013) Science paper already included a graph showing the failure of all CMIP5 models to reproduce the percentage amplification of the NHL seasonal cycle.... It would be most helpful if the authors could shed some more light on this issue, and important that they should explain why their conclusion appears to differ from that of Graven et al. (2013).*

- Thomas et al. (2016) used the MsTMIP ensemble, i.e., only the land components of ESMs driven by observed forcings. Graven et al. (2013) analyzed CMIP5 simulations tagged *historical*, i.e., CO₂ concentration forced simulations (Taylor et al., 2012).
- Simulations by Earth system models may be either concentration driven or emission driven. The former are directly driven by time-evolving CO₂ concentration and emissions are implicit, and the latter are forced by time-evolving emissions of CO₂ and the concentration is simulated for different layers of the atmosphere (Boer and Arora, 2012).
- By using CO₂ concentration driven simulations, Thomas et al. and Graven et al. analyzed models that are uncoupled in terms of the carbon cycle. This neglects feedbacks between interacting terrestrial and oceanic biogeochemical components and the atmosphere (Boer and Arora, 2012). Both studies used atmospheric transport models to translate the carbon flux output of land models into three-dimensional fields of CO₂ concentration. It is also possible that a systematic bias in the transport models causes the ubiquitous underestimation in the reproduction of the CO₂ seasonal cycle.
- Wenzel et al. (2016) and our research used a different set of models and simulations. The models used for the CMIP5 simulations, tagged *esmHistorical*, are CO₂ emission driven and, thus, are ESMs with a fully-coupled global carbon cycle. In this setup, ocean and land carbon components are linked via the transport of CO₂ in the atmosphere in each ESM. As a consequence, these are different models with different characteristics (e.g. varying radiative and CO₂ fertilization effects) and equilibria than the ones used by Graven et al. Boer and Arora (2012) looked into the differences in the carbon cycle behavior of an ESM, CanESM2, for emission vs. concentration driven simulations.
- Wenzel et al. (2016) also showed that in the CMIP5 *esmHistorical* ensemble, most models are underestimating the change in the NHL CO₂ seasonal cycle, but not all, as concluded by Thomas et al. and Graven et al.

- Moreover, Graven et al. and Thomas et al. investigated relative changes of the CO₂ amplitude and concluded that the models underestimate these. We looked into absolute changes. The model skill of reproducing relative and absolute changes are not connected.

2. The authors cite the flawed study of Kolby Smith and others apparently showing that models over-estimate the CO₂ fertilization effect. Either this reference should be deleted, or the subsequent correspondence item also in NCC debunking Smith et al. should also be cited. (I declare an interest as a co-author of the correspondence item, but I believe my point is valid nonetheless!)

- We agree with the referee and included the correspondence reference as suggested (Page 2, Line 8; Page 8, Lines 26-28).

3. It seems to me that the authors may have missed an opportunity to make separate estimates of the physiological and climate effects. They are not separable in terms of the overall trend, however the fact that climate shows strong interannual variability while CO₂ show a monotonically increasing trend should allow the different effects to be separated statistically.

- A similar analysis has also been conducted, which allows a rather clean separation of physiological and climate effects. The focus of the manuscript in hand, however, is not the factorial driver separation of photosynthesis enhancement and, thus, the results were not included. Currently, we are drafting another manuscript covering this subject matter in depth.

References:

Boer, G. J., & Arora, V. K. (2012). Feedbacks in Emission-Driven and Concentration-Driven Global Carbon Budgets. *Journal of Climate*, 26(10), 3326–3341.

Graven, H. D., Keeling, R. F., Piper, S. C., Patra, P. K., Stephens, B. B., Wofsy, S. C., ... Bent, J. D. (2013). Enhanced Seasonal Exchange of CO₂ by Northern Ecosystems Since 1960. *Science*, 341(6150), 1085–1089.

Taylor, K. E., Stouffer, R. J., & Meehl, G. A. (2012). An Overview of Cmp5 and the Experiment Design. *Bulletin of the American Meteorological Society*, 93(4), 485–498.

Thomas, R. T., Prentice, I. C., Graven, H., Ciais, P., Fisher, J. B., Hayes, D. J., ... Zeng, N. (2016). Increased light-use efficiency in northern terrestrial ecosystems indicated by CO₂ and greening observations. *Geophysical Research Letters*, 43(21), 11,339-11,349.

Wenzel, S., Cox, P. M., Eyring, V., & Friedlingstein, P. (2016). Projected land photosynthesis constrained by changes in the seasonal cycle of atmospheric CO₂. *Nature*, 538(7626), 499–501.

Reviewers' comments:

Reviewer #1 (Remarks to the Author):

With all due respect, I find the author's responses to my comments regarding the applicability of the Emerging Constraint (EC) approach to high-latitude vegetation responses over the 21st century using CMIP5 models to be fundamentally flawed. The literature I cite clearly demonstrates that the CMIP5 models did not credibly represent many of the dominant controllers on high-latitude vegetation: (1) responses to warming and increased nutrient availability; (2) soil temperatures (which affect soil biogeochemistry and nutrient availability); (3) light competition; and (4) atmospheric CO₂. The authors' contentions that they are examining above ground responses, and that therefore CMIP5 models' extremely poor (or non-existent) representations of the dominant controllers on those responses, is clearly a deficient defense of their approach.

The authors state that "Four out of seven models in our study represent nutrient constraints" is false if they used the CMIP5 models. Only CESM1-BGC and NorESM-ME included explicit nitrogen dynamics and constraints in the CMIP5 archive, and those ESMs used the same land model (CLM4). CLM4, as I described in my first review, has been shown over the past several years to be fundamentally flawed conceptually and in comparison with observations (particularly at high latitudes).

I appreciate the authors' efforts to examine the FLUXCOM inferences of GPP and their relationships with satellite-derived LAI. Perhaps that analysis will be helpful when the CMIP6 archive becomes available, which should include many more models with realistic high-latitude vegetation responses to climate, CO₂, and soil BGC.

I note that I think the EC approach can be valuable, but, in my opinion, the underlying models must at least realistically represent some reasonable fraction of the processes responsible for the predicted response. That criterion was not met in the CMIP5 archive at any level for high-latitude vegetation responses to climate change.

Reviewer #2 (Remarks to the Author):

Reviewed by Matthias Forkel, 2018-06-18

I thank Winkler et al. for their good response to the referee comments and the appropriate changes in the manuscript. I think that the manuscript is now much better and almost ready to be published. Before publication, the authors should still improve the following minor aspects.

-- Figure 3 a+b: The amplitude is usually the difference between the seasonal maximum and minimum CO₂. Hence the y-axis label is misleading. "CO₂ amplitude" should be replaced with "Detrended CO₂" or "CO₂ seasonal cycle"

-- Figure 3 d: Why are results shown for MPI-ESM in Fig. 3 d, which is however not shown in Fig. 3 a and b? Is it possible to show in Fig. 3 d results from a model that is also used in Fig. 3 a and b?

-- "slope of summertime drawdown" (page 6, line 6, Figure 3 c, and other places)

You need to better explain the term "drawdown slope". I do not really understand from your description what this term actually means. I understand this term as the slope between the CO₂ maximum in spring and the CO₂ minimum in late summer, i.e. the annual rate of CO₂ drawdown. I see this meaning also reflected in what you write in the main text (page 6). However, when I read the methods (§5), I get the impression that you calculated a linear regression between the detrended CO₂ from June/July across all years, which would be simply a temporal trend in the

seasonal CO₂ minima. Can you please clarify and better describe what exactly the “drawdown slope” is and how it was calculated.

-- Estimated change in NPP (page 6, lines 26-30 to page 7 lines 1-4)

I assume that the reported values of NPP (and NBP) changes are differences between the mean NPP (NBP) in the periods 2000-2015 and 1980-1985. Is this correct? If this is the case, I think that the change in NPP (0.32 PgC yr⁻¹) from the EC-based translation cannot be directly compared to the change in NBP from the JENA estimate and that it is an underestimation. This comparison assumes that the NBP change comes only from NPP and that heterotrophic respiration and disturbances did not change. As the authors correctly write, heterotrophic respiration may have increased as well and hence the NPP increase needs to be much larger to explain the positive change in NBP. For example, the LPJmL model simulated in boreal and arctic regions over 1970-2011 a GPP increase of 0.065 PgC yr⁻² (* 42 years = 2.73 PgC absolute change) and a Reco increase of 0.061 PgC yr⁻² (= 2.56 PgC) (Fig. S8 in (Forkel et al., 2016)). This corresponds to an NBP increase of 0.011 PgC yr⁻² (= 0.352 PgC) in 1980-2011 which is consistent with the JENA NBP estimate (Fig. S10 in (Forkel et al., 2016)). Given that the LPJmL model reproduces greening and CO₂ amplitude trends, the simulated changes in GPP and Reco are likely plausible and indicate that the change in NPP needs to be much higher (i.e. GPP trend = 0.065 PgC yr⁻² * 30 years = 1.95 PgC \diamond if we assume that NPP/GPP is constant \sim 0.5, the NPP trend would be \sim 0.98 PgC). I partly agree with the concluding sentence of the paragraph that “evidence from inversion studies of atmospheric CO₂ measurements indicates NPP changes in NHL comparable to or larger than our EC estimate, and therefore the multi-model mean to be an underestimate.” So I think that the entire paragraph needs to be rewritten in a way that the EC-based NPP change represents a minimum estimate (not a real estimate of NPP changes) under the assumption that heterotrophic respiration and fires did not change. Hence as the models already underestimate the minimum estimate, the actual change will be also underestimated. Actually, I find your answer to my comment more understandable and logical than the text in the manuscript:

“We derive an estimate for how much NPP must have increased to explain this decrease in CO₂ minimum. We find that it points to the same direction as the EC estimate and buttresses the statement that NPP increase might be underestimated in most models. Now, increase in heterotrophic respiration (Rh) during the growing season (e.g. due to warming) would result in an increase of the annual CO₂ minimum. If this is the case, then NPP must have increased even more to counterbalance the increase in Rh. Thus, the derived value of NPP enhancement is a conservative estimate in comparison with the NBP estimates by the inversion products (also because fire emissions etc. are not considered). We do not claim that this method provides an estimate for actual NPP increase in the northern high latitudes. We rather show that a simple back-of-the-envelope calculation based on long-term measurements of CO₂ concentration yields an estimate that is in the same ballpark as the greening-based Emergent Constraint value.”

Authors' Response to Reviewer 1 (NCOMMS-17-32151A)

We thank the Reviewer for reviewing the revised manuscript. However, we find that his / her comments to be inaccurate and false in some cases – we include all comments of his / her 1. Review at the end of this document to contest this Reviewer's comments.

Comments in the 2. Review

1. With all due respect, I find the author's responses to my comments regarding the applicability of the Emerging Constraint (EC) approach to high-latitude vegetation responses over the 21st century using CMIP5 models to be fundamentally flawed. The literature I cite clearly demonstrates that the CMIP5 models did not credibly represent many of the dominant controllers on high-latitude vegetation: (1) responses to warming and increased nutrient availability; (2) soil temperatures (which affect soil biogeochemistry and nutrient availability); (3) light competition; and (4) atmospheric CO₂.

We respectfully reject the Reviewer's statement that our responses to his / her comments are fundamentally flawed. We addressed every comment with in-depth explanations, additional analysis, and reviewed as well as discussed all six studies cited by the Reviewer.

All CMIP5 models include biogeochemical response and radiative response (warming of air and soils) of high-latitude vegetation to rising atmospheric CO₂ concentration. Four out of seven models integrate nutrient limitation using different approaches (NorESM1-ME, CESM1-BGC, CanESM2, and MIROC-ESM). A different set of four models represent vegetation dynamics also including explicit or implicit light competition (MPI-ESM-LR, GFDL-ESM2M, HadGEM2-ES, MIROC-ESM). The characteristics of the individual models used in this study are described in the manuscript (e.g. ED Extended Data Table 2) and are well documented (**MPI-ESM-LR**: Raddatz et al., *Clim. Dyn.*, 2007; Reick et al., *J. Adv. Model. Earth Syst.*, 2013; **CanESM2**: Arora et al., *Geophys. Res. Lett.*, 2011; **MIROC-ESM**: Watanabe et al., *Geosci. Model Dev.*, 2011; **NorESM1-ME**: Bentsen et al., *Geosci. Model Dev.*, 2013; **CESM1-BGC**: Lindsay et al., *J. Climate*, 2014; **GFDL-ESM2M**: Dunne et al., *J. Climate*, 2012; **HadGEM2-ES**: Collins et al., *Geosci. Model Dev.*, 2011).

The models' representation of high-latitude vegetation response to changing dominant controllers results in a large spread of simulated gross primary production (GPP) and greening (Fig. 2a, b, c; discussed multiple times in the manuscript). For this very reason, the method of Emergent Constraints was developed and applied here. The concept relies on the collective agreement of the multi-model ensemble on the essential relationship between the predictor (in our case greening sensitivity) and the predictand (in our case GPP enhancement). The models individually can be wrong with respect to observations, but together they can engender a better understanding of the system.

The Reviewer states that ESMs using the land component CLM4 are flawed and are not reproducing observations. In Figure R1 below, we present the original EC figure (Fig. 2c in the manuscript) and the EC relationship excluding the CLM4-based models CESM1-BGC and NorESM1-ME. The EC estimate is nearly the same. This is the key aspect of the EC approach. In comparison to the conventional model average value, the EC estimate is not susceptible to individual models, but relies on the collective behavior of all models.

This functionality of the EC method has been demonstrated in numerous studies. For example, in a study by Lian et al. (*Nature Climate Change*, 2018) published in July this year. They present

a CMIP5 based EC estimate of the global ratio of plant transpiration to evapotranspiration using site-level data. None of the CMIP5 models reproduces the observations and all models underestimate the observed values. However, in the ensemble it becomes evident that models that vastly underestimate at the site-level, also underestimate on global scale. The resulting linear relationship between site-level and global estimates across the ensemble renders it possible to use the site-level data to derive an observational-based global estimate.

Figure R1.1 | **a**, Emergent Constraint based on the complete CMIP5 ensemble. **b**, Emergent Constraint excluding CLM4-based models CESM1-BGC and NorESM1-ME.

2. The authors' contentions that they are examining above ground responses, and that therefore CMIP5 models' extremely poor (or non-existent) representations of the dominant controllers on those responses, is clearly a deficient defense of their approach.

In our responses to the first round of comments, we included a short review of all six studies cited by the Reviewer. Only two (Koven et al., J. Climate, 2012; Bouskill et al., Biogeosciences, 2014) are focused on high-latitude processes, but are concerned with soil carbon dynamics. We emphasized that our study relies on aboveground processes, so GPP and associated LAI response to rising atmospheric CO₂ concentration and increasing air temperature. While there are studies that show CMIP5 models poorly match observations in certain specific cases, this is not a reason to discount the EC approach. We argue that this very disagreement between models and data is the basis of EC method's legitimacy. Also see response to Comment 1.

3. The authors state that "Four out of seven models in our study represent nutrient constraints" is false if they used the CMIP5 models. Only CESM1-BGC and NorESM-ME included explicit nitrogen dynamics and constraints in the CMIP5 archive, and those ESMS used the same land model (CLM4).

First, our complete sentence reads "Four out of seven models in our study represent nutrients constraint, where CESM1-BGC and NorESM1-ME include an explicit nitrogen cycle (Page 7, Lines 6-16)" (Review 1, Response to Comment 1.3). We explicitly stated that the models CESM1-BGC and NorESM1-ME include an explicit nitrogen cycle, in the response to the reviewer as well as in the manuscript. Reviewer 1 only quotes the first part of our sentence,

thus neglecting the second part, and then uses this bit of information to claim we made a false statement.

Second, our statement is not false. In fact, four models in the CMIP5 ensemble (NorESM1-ME, CESM1-BGC, CanESM2, and MIROC-ESM) represent nutrient constraints. CESM1-BGC and NorESM1-ME integrate an explicit nitrogen cycle, MIROC-ESM and CanESM2 rely on an empirical approach to include nitrogen limitation. This aspect is already discussed in more detail in the manuscript (Page 7, Lines 6-16) and in the cited article of Arora et al. (2013) on Carbon-Climate Dynamics in CMIP5, under section ‘4. Feedback Contributions’: “In the absence of an explicit terrestrial nitrogen cycle, the strength of the CO₂ fertilization effect in CanESM2 is “downregulated” based on the response of plants grown in ambient and elevated CO₂ [...]” and “[...] MIROC-ESM uses an empirical approach to model the photosynthetic response to CO₂ which implicitly includes the response to nutrient limitation.” (Arora et al., 2013).

It seems to us that Reviewer 1 did not check the references nor did she / he read the associated section in the revised manuscript.

4. CLM4, as I described in my first review, has been shown over the past several years to be fundamentally flawed conceptually and in comparison with observations (particularly at high latitudes).

The reviewer did not describe nor even mention the CLM4 model in his / her 1. Review. On the contrary, Reviewer 1 claimed that none of the tested models represent nutrient constraints (Review 1, Comment 1.3). We responded and clarified that Reviewer 1 is mistaken and revised the manuscript to be more clear on that matter.

5. I appreciate the authors’ efforts to examine the FLUXCOM inferences of GPP and their relationships with satellite-derived LAI. Perhaps that analysis will be helpful when the CMIP6 archive becomes available, which should include many more models with realistic high-latitude vegetation responses to climate, CO₂, and soil BGC. I note that I think the EC approach can be valuable, but, in my opinion, the underlying models must at least realistically represent some reasonable fraction of the processes responsible for the predicted response. That criterion was not met in the CMIP5 archive at any level for high-latitude vegetation responses to climate change.

As explained in response to Comment 1, the EC approach does not demand models that realistically reproduce observations. In the EC literature, it has been shown that the relationship between predictor and predictand does not change qualitatively using an ensemble of updated models (Wenzel et al., J. Geophys. Res. Biogeosci., 2014 vs. Cox et al., Nature, 2013; and Hall and Qu, Geophys. Res. Lett., 2006 vs. Qu and Hall, Clim. Dyn., 2014). In other words, models that simulate strong GPP enhancement will also simulate strong greening in CMIP6. Ideally for CMIP6, the inter-model spread will decrease, but that has no effect on the EC estimate itself. For example, this aspect has been shown in two studies constraining projections of the snow-albedo feedback in the northern high-latitudes. In 2006, Hall and Qu presented an EC estimate based on large inter-model variations in snow-albedo feedback strength of the current seasonal cycle and under climate change using CMIP3. In 2014, Qu and Hall confirmed the CMIP3-based findings redoing the analysis for the CMIP5 ensemble. The authors also showed that the agreement between models and observations did not improve in the updated ensemble.

Comments in the 1. Review

Winkler et al. apply an Emerging Constraint (EC) approach by combining recent climate and satellite LAI observations to estimate GPP sensitivity to forcing. This sensitivity is then extrapolated to the future to conclude that CMIP5 ESMs underestimate the land's ability to assimilate atmospheric CO₂. The EC approach is interesting, but I agree with the previous reviewers that its application must be more carefully considered. I think the authors have not sufficiently demonstrated that they have met the basic assumptions underlying the EC approach, for several reasons (some of which were raised by previous reviewers):

1.1 The Emerging Constraint (EC) approach requires several assumptions be satisfied, e.g. the models used for extrapolation must believably represent the majority of processes responsible for system dynamics over the current and extrapolated time frames.

1.2 The CMIP5 models used in this study have been analyzed in many studies, and they have been shown to either poorly represent, or not even include, the dominant processes important for high-latitude hydrological, geomorphological, coupled carbon and nutrient, thermal, or species distribution changes (Bouskill et al., 2014; Todd-Brown et al., 2013; He et al., 2016; Koven et al., 2013; De Kauwe et al., 2017; Zaehle et al., 2014).

1.3 For example, given that high-latitude systems (1) are often strongly N limited; (2) CO₂ fertilization requires additional N supply; and (3) none of the tested models even represent nutrient constraints, it is hard to believe they are even predicting the current constraints on GPP reasonably, much less the future responses.

Given this, and other, evidence, I do not find it believable that the CMIP5 simulations can be used to infer high-latitude GPP sensitivity to climate over decades.

2.1 That extrapolation must be linear with respect to the independent variable being used (i.e., ?). Although the models show this linearity, there is no demonstration that the real world follows a linear relationship. Current understanding of high-latitude system dynamics indicates that these ecosystems will respond non-linearly to potentially dramatic changes in geomorphology, hydrology, plant distributions, fire, and nutrient availability.

2.2 Given that the CMIP5 models clearly have unrealistic representations of high-latitude processes, using them to make an argument about linearity seems inappropriate.

3.1 given all the complexity of linking changes in NDVI to changes in LAI to changes in GPP, especially for high latitude systems;

3.2 no assessment of the uncertainties in this calculation; and

3.3 no demonstration using any ground observations to test whether the hypothesized links between GPP and are accurate, it does not seem reasonable to conclude that the calculated GPP versus relationship is sufficiently robust to justify an extrapolation to the future.

Authors Response to Reviewer 2 Dr. M. Forkel (NCOMMS-17-32151A)

I thank Winkler et al. for their good response to the referee comments and the appropriate changes in the manuscript. I think that the manuscript is now much better and almost ready to be published. Before publication, the authors should still improve the following minor aspects.

We thank Dr. Forkel for recommending the manuscript to be published. We also thank for his in-depth and constructive comments on the revised manuscript. The revisions done in response to his four minor comments has resulted in an improved manuscript.

1. Figure 3 a+b: The amplitude is usually the difference between the seasonal maximum and minimum CO₂. Hence the y-axis label is misleading. “CO₂ amplitude” should be replaced with “Detrended CO₂” or “CO₂ seasonal cycle”

We agree that the “CO₂ amplitude” as y-label is misleading. We modified Figure 3a+b as suggested by the Reviewer.

2. Figure 3 d: Why are results shown for MPI-ESM in Fig. 3 d, which is however not shown in Fig. 3 a and b? Is it possible to show in Fig. 3 d results from a model that is also used in Fig. 3 a and b?

Unfortunately, we only have recourse to high-resolution simulations (ensemble of 10 realizations) of the MPI-ESM. High spatial resolution and multiple ensemble members are needed to investigate on the seasonality of the Arctic Ocean. Fig. 3d shows that the northern lands are responsible for changes in CO₂ amplitude. The different models most likely vary in absolute terms, but qualitatively they should agree on the dominance of terrestrial processes in the high latitudes.

**3. “slope of summertime drawdown” (page 6, line 6, Figure 3 c, and other places)
You need to better explain the term “drawdown slope”. I do not really understand from your description what this term actually means. I understand this term as the slope between the CO₂ maximum in spring and the CO₂ minimum in late summer, i.e. the annual rate of CO₂ drawdown. I see this meaning also reflected in what you write in the main text (page 6). However, when I read the methods (§5), I get the impression that you calculated a linear regression between the detrended CO₂ from June/July across all years, which would be simply a temporal trend in the seasonal CO₂ minima. Can you please clarify and better describe what exactly the “drawdown slope” is and how it was calculated.**

As the reviewer correctly stated, the drawdown slope is the slope between the CO₂ maximum in spring and the CO₂ minimum in late summer. We modified the Methods section accordingly (Page 17, Lines 12-20).

**4. Estimated change in NPP (page 6, lines 26-30 to page 7 lines 1-4)
I assume that the reported values of NPP (and NBP) changes are differences between the mean NPP (NBP) in the periods 2000-2015 and 1980-1985. Is this correct? If this is the case, I think that the change in NPP (0.32 PgC yr⁻¹) from the EC-based translation cannot**

be directly compared to the change in NBP from the JENA estimate and that it is an underestimation . This comparison assumes that the NBP change comes only from NPP and that heterotrophic respiration and disturbances did not change. As the authors correctly write, heterotrophic respiration may have increased as well and hence the NPP increase needs to be much larger to explain the positive change in NBP. For example, the LPJmL model simulated in boreal and arctic regions over 1970-2011 a GPP increase of 0.065 PgC yr-2 (42 years = 2.73 PgC absolute change) and a Reco increase of 0.061 PgC yr-2 (= 2.56 PgC) (Fig. S8 in (Forkel et al., 2016)). This corresponds to an NBP increase of 0.011 PgC yr-2 (= 0.352 PgC) in 1980-2011*

*which is consistent with the JENA NBP estimate (Fig. S10 in (Forkel et al., 2016)). Given that the LPJmL model reproduces greening and CO2 amplitude trends, the simulated changes in GPP and Reco are likely plausible and indicate that the change in NPP needs to be much higher (i.e. GPP trend = 0.065 PgC yr-2 * 30 years = 1.95 PgC, if we assume that NPP/GPP is constant ~ 0.5, the NPP trend would be ~0.98 PgC). I partly agree with the concluding sentence of the paragraph that “evidence from inversion studies of atmospheric CO2 measurements indicates NPP changes in NHL comparable to or larger than our EC estimate, and therefore the multi-model mean to be an underestimate.” So I think that the entire paragraph needs to be rewritten in a way that the EC-based NPP change represents a minimum estimate (not a real estimate of NPP changes) under the assumption that heterotrophic respiration and fires did not change. Hence as the models already underestimate the minimum estimate, the*

actual change will be also underestimated. Actually, I find your answer to my comment more understandable and logical than the text in the manuscript:

“We derive an estimate for how much NPP must have increased to explain this decrease in CO2 minimum. We find that it points to the same direction as the EC estimate and buttresses the statement that NPP increase might be underestimated in most models. Now, increase in heterotrophic respiration (Rh) during the growing season (e.g. due to warming) would result in an increase of the annual CO2 minimum. If this is the case, then NPP must have increased even more to counterbalance the increase in Rh. Thus, the derived value of NPP enhancement is a conservative estimate in comparison with the NBP estimates by the inversion products (also because fire emissions etc. are not considered). We do not claim that this method provides an estimate for actual NPP increase in the northern high latitudes. We rather show that a simple back-of-the-envelope calculation based on long-term measurements of CO2 concentration yields an estimate that is in the same ballpark as the greening-based Emergent Constraint value.”

Based on the suggestion, we rewrote this section to be more clear on how the different estimates of NPP and NBP are put into comparison (Page 6, Line 21 – Page 7, Line 5)

REVIEWERS' COMMENTS:

Reviewer #2 (Remarks to the Author):

The authors sufficiently responded to my previous comments. I have no further remarks.

I noticed the strong disagreements between reviewer 1 and the authors. I agree with reviewer 1 that many processes of vegetation and carbon cycle dynamics in high latitude regions are not well or not at all represented in CMIP5 models. In my view, the authors provided, however, a good and sufficient response to the reviewer comments in both rounds of the review. The authors clearly state the limitations of CMIP5 models in their response letters and in the manuscript (page 7, line 7-17).